# Structural studies of the IFNλ4 receptor complex using cryoEM enabled by protein engineering

William S. Grubbe[1,6], Bixia Zhang[2,6], Aileen Kauffman [1], Fabian Byléhn [1], Kasia Padoł[1], Hae-Gwang Jung[3], Seung Bum Park [3], Jessica M. Priest [2], Engin Özkan [2], Juan J. de Pablo [1,4], T. Jake Liang [3], Minglei Zhao [2] & Juan L. Mendoza [1,2,5] ✉

IFNλ4 has posed a conundrum in human immunology since its discovery in 2013, with its expression linked to complications with viral clearance. While genetic and cellular studies revealed the detrimental effects of IFNλ4 expression, extensive structural and functional characterization has been limited by the inability to express and purify the protein, complicating explanations of its paradoxical behavior. In this work, we report a method for robust production of IFNλ4. We then use yeast surface display to affinity-mature IL10Rβ and solve the 72 kilodalton structures of IFNλ4 (3.26 Å) and IFNλ3 (3.00 Å) in complex with their receptors IFNλR1 and IL10Rβ using cryogenic electron microscopy. Comparison of the structures highlights differences in receptor engagement and reveals a distinct 12-degree rotation in overall receptor geometry, providing a potential mechanistic explanation for differences in cell signaling, downstream gene induction, and antiviral activities. Further, we perform a structural analysis using molecular modeling and simulation to identify a unique region of IFNλ4 that, when replaced, enables secretion of the protein from cells. These findings provide a structural and functional understanding of the IFNλ4 protein and enable future comprehensive studies towards correcting IFNλ4 dysfunction in large populations of affected patients.

When they were initially discovered, type III interferons (IFNλ1-3) were structurally and evolutionarily classified as members of the IL-10 superfamily of cytokines. Further investigation, though, revealed that they are functionally interferons (IFNs) with distinct roles in mucosal and innate immunity[1–3]. In contrast to other IFNs, expression of one of their receptors, IFNλR1, is tissue-restricted, motivating studies to leverage this potential therapeutic property. However, the clinical use of IFNs in general has been limited for reasons ranging from side effects associated with off-target signaling (type I IFNs) to lack of

efficacy (type III IFNs)[4,5]. Importantly, key differences in patient genotypes are strongly associated with differences in IFN treatment response[6,7] and implicate the IFNλ genetic locus, specifically the expression of a newly identified protein called IFNλ4, as a culprit.

Since its discovery in 2013, IFNλ4 has perplexed researchers due to the paradoxical observation that in vivo expression leads to lower viral clearance, but that limited and poorly quantified in vitro data suggests antiviral activity comparable to other IFNλs[6,8–10]. The global and public health implications of expressing IFNλ4 are significant, as

[1]Pritzker School of Molecular Engineering, University of Chicago, Chicago, IL, USA. [2]Department of Biochemistry and Molecular Biology, University of Chicago, Chicago, IL, USA. [3]Liver Diseases Branch, National Institute of Diabetes and Digestive and Kidney Diseases (NIDDK), National Institutes of Health, Bethesda, MD, USA. [4]Argonne National Laboratory, Lemont, IL, USA. [5]Howard Hughes Medical Institute, University of Chicago, Chicago, IL, USA. [6]These authors contributed equally: William S. Grubbe, Bixia Zhang. ✉e-mail: jlmendoza@uchicago.edu

increasing numbers of studies negatively link IFNλ4 expression to disease presentation and severity ranging from Hepatitis C to COVID-19[6,11–14]. With surprising amounts of the world population predicted to express this protein (as high as 70.7% of people of African descent, 31.2% European, and 8% Asian)[15], understanding the structural and functional properties of the IFNλ4 protein is critical to elucidating its role in the human immune system, unraveling this immunological conundrum, and developing methods for effective patient treatment.

It is known that IFNλ4 is poorly secreted in both natural and recombinant systems and that its expression induces cell stress via intracellular accumulation[16,17]. However, the extracellular function of IFNλ4 through its engagement of IL10Rβ and IFNλR1 remains poorly understood. While predicted to be structurally similar to IFNλ1-3, IFNλ4 shares only 28% amino acid sequence identity with its closest relative IFNλ3 and has a distinct biochemical composition resulting in IFNλ4 being one of the most positively-charged known cytokines (pI = 11.3). These differences evoke intriguing structural questions regarding potential differences in receptor engagement between IFNλ3 and IFNλ4 and their resulting influences on cell signaling and IFN function. Functional comparisons of these proteins have become increasingly relevant as their differential expression and associated single nucleotide polymorphisms have been shown to be important for viral progression in patients[8,18–21]. While cellular and genetic investigations comparing the two proteins are relatively more extensive, fewer studies have focused on their extracellular functions, and none have investigated a structural basis for these differences observed in vitro and in vivo. This is largely due to well-reported difficulties expressing and purifying IFNλ4 and the resulting inability to produce the quantities of protein needed for structural and quantitative functional studies. Indeed, methods to express small quantities, as well as engineered versions of IFNλ4 protein, have been reported[22,23], but a method for robust expression of unmodified IFNλ4 has not. To answer these remaining structural and functional questions surrounding IFNλ4, improved methods for its expression and purification are essential.

In this study, we establish a method for high-yield, unmodified IFNλ4 expression and purification to resolve this bottleneck and enable structural and highly quantitative functional studies of IFNλ4. We next use protein engineering to overcome the affinity limitation for structural determination of the ternary IFNλ cytokine-receptor complex and determine the 72 kilodalton (kDa) structures of IFNλ4 as well as IFNλ3 in complex with IFNλR1 and an affinity-matured IL10Rβ via cryogenic electron microscopy (cryoEM). Analysis of these structures elucidates the mechanism of extracellular engagement by IFNλ4 with its receptors, highlighting distinct differences in residue engagement and receptor geometry relative to IFNλ3. We elevate our understanding of these complexes using molecular dynamics simulations and identify a disordered and non-receptor-engaging region of IFNλ4, lending a potential structural explanation to its poor secretion. Lastly, with our improved method for protein expression, we perform quantitative investigations of the extracellular function of IFNλ4 and identify kinetic relationships between gene activation and viral clearance for the IFNλ3 and IFNλ4 proteins, with these differences in vitro potentially explained by the structural differences in the receptor complexes. In total, this study paints a comprehensive portrait of the IFNλ4 protein, revealing the molecular details of its function and enabling rigorous in vitro and in vivo characterization towards overcoming the therapeutic barrier of IFNλ4 expression in afflicted patient populations.

## Results

### Robust expression and purification of unmodified IFNλ4
Issues with expression and isolation of the IFNλ4 protein have been consistently reported in the field and presented a significant bottleneck to structural and functional studies. Bacterial expression

followed by refolding of the protein has been the standard protocol for many studies, while some approaches have taken advantage of glycoengineering of the protein to increase apparent solubility and expression. Regardless of the method, however, the yield of protein has been too low to consider pursuing structural studies, and many of these previously reported methods have lacked quantification of the final product. We developed a protein expression construct and purification protocol to enable cellular secretion of IFNλ4 by covalently linking it via a flexible peptide linker containing a protease recognition site to its low-affinity receptor, IL10Rβ[24]. To increase the molecular weight of the construct and achieve complete separation of IFNλ4 during fast-protein liquid chromatography (FPLC), we added sfGFP protein to the N-terminus of IL10Rβ (Fig. 1A) such that pure IFNλ4 could be recovered (Fig. 1B, Supplementary Fig. 1). Using this method, yields of more than 1 mg/L of total protein were achieved with an insect cell expression system. Increasing the concentration of sodium chloride to 500 millimolar (mM) at all steps of protein harvest and purification was essential due to the sticky nature of IFNλ4, reflecting its high pI.

### Engineering a high-affinity IL10Rβ for the solution of type III IFN ternary complexes
After resolving the IFNλ4 expression bottleneck, we turned our attention to solving the structure of the protein in complex with its two receptors, IFNλR1 and IL10Rβ. Like other protein complexes involving IL10Rβ, the native affinity of IL10Rβ for the IFNλ complex is too low to enable isolation of the ternary complex for structural studies. Previous attempts to overcome this limitation leveraged directed evolution to engineer a high-affinity ligand to enhance overall complex affinity[10,25,26]; however, this approach is not viable for IFNλ4, as the protein does not display for presumably the same reasons limiting its expression. Thus, we elected to perform directed evolution on IL10Rβ to improve the binding of the low-affinity receptor and enable the solution of the IFNλ3 and IFNλ4 ternary complexes, allowing for direct comparisons of the two ligands solved using the same methods and conditions.

We used yeast surface display to engineer a high-affinity IL10Rβ. A library of $5 \times 10^7$ clones was generated using error-prone polymerase chain reaction (PCR), and yeast displaying IL10Rβ were subjected to multiple rounds of selection (Fig. 1C) as previously described[27,28]. The selection strategy involved first performing four rounds of selection with IFNλ3 as the ligand and decreasing concentrations of biotinylated IFNλR1 as the selection pressure was to intentionally select for versions of IL10Rβ with increased affinity at the receptor-receptor interface. The final two rounds of selection used refolded IFNλ4 as a ligand to ensure that the affinity-enhanced receptor would recognize these related but sequence-dissimilar proteins. Ninety-six clones from this final selection were screened for binding to IFNλ4 and IFNλR1, and the highest affinity clone (referred to as A3) was selected after demonstrating improved complex formation on-yeast relative to the wild-type IL10Rβ for both IFNλ4 and IFNλ3 (Fig. 1D, Supplementary Fig. 2). Genetic sequencing of A3 revealed three mutations at residues 147, 148, and 184, with residues 147 and 148 located on Loop 5 of IL10Rβ near the ternary complex junction (Fig. 1E). The mutation at residue 147 was the same mutation (N147D) identified by molecular dynamics simulations previously used to inform the design of a type III IFN complex with enhanced ternary complex stability via increased protein-protein interactions (PPIs)[29]. In line with the yeast data, IL10Rβ-A3 was able to form a stable ternary complex by gel filtration chromatography with both IFNλ3:IFNλR1 and IFNλ4:IFNλR1, enabling structural studies (Supplementary Fig. 3).

### Structures of the IFNλ4 and IFNλ3 receptor ternary complexes
To facilitate structural determination, an N-glycan minimized version of the IL10Rβ-A3 protein was used as previously described[27].

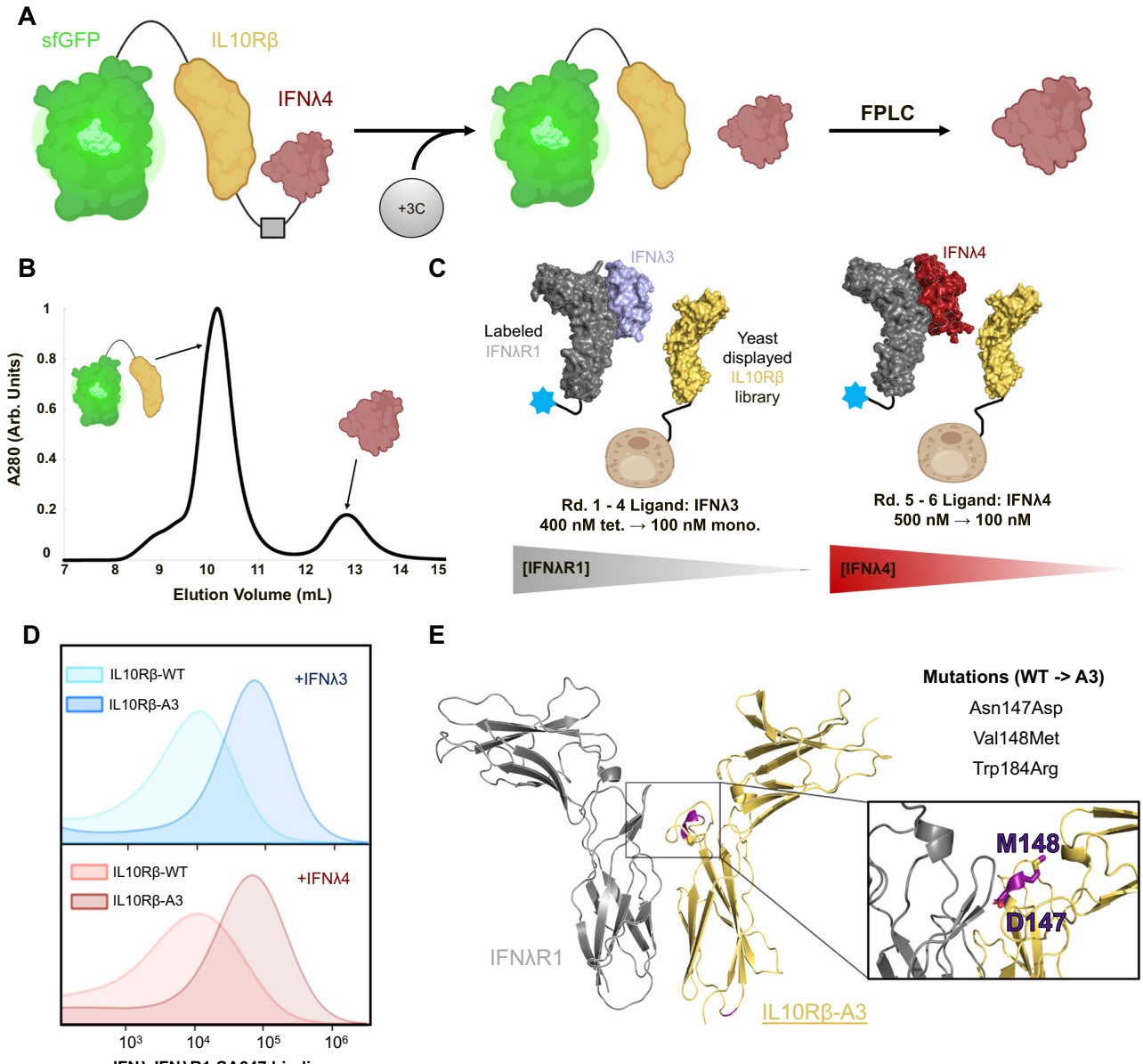

**Fig. 1 | Robust expression of IFNλ4 and protein engineering of IL10Rβ enable structural studies of the ternary receptor complex. A** Expression of IFNλ4 is enabled by covalently linking it to IL10Rβ via a protease-cleavable site. Following treatment, IFNλ4 is harvested and purified via fast-protein liquid chromatography (FPLC). **B** FPLC trace of sfGFP-10RB-IFNλ4 after treatment with 3C protease (Arb. Units = arbitrary units). **C** Wild-type IL10Rβ was displayed on yeast and subjected to six rounds of selection to increase the overall affinity of the ternary protein complex. (Round 1–4: [IFNλR1] from 400 nM tetramers to 100 nM monomer. Round 5–6: [IFNλ4] from 500 nM to 100 nM). **D** Yeast displaying either IL10Rβ (WT) or IL10Rβ-A3 (A3) stained with either 1 μM IFNλ3:IFNλR1 (top, blue) or IFNλ4:IFNλR1 (bottom, red) (SA647 = streptavidin-Alexa Fluor 647). **E** Mutations N147D and V148M are located at the interface of the three proteins (inset, in purple). W184R is located towards the C-terminus of IL10Rβ-A3 [(**A**, **B**) were created, in part, in BioRender. Grubbe, W. (2024) https://BioRender.com/i17x353].

Previous structural studies of related complexes relied on X-ray crystallography[27,30]; however, the lower yield of IFNλ4 relative to IFNλ3, its tendency to stick and aggregate during concentration, and the conditions required to purify it became impassable barriers for large-scale crystallization trials. The ternary complexes of IFNλs with their receptors are around 72 kDa, making them challenging but exciting samples for use in single-particle cryoEM[31,32]. Through extensive optimization of sample preparation, we successfully determined both ternary complexes using single-particle analysis to final resolutions of 3.26 Å for IFNλ4 (Fig. 2A) and 3.00 Å for IFNλ3 (Fig. 2B). In both data sets, ~5% of particles from the initial particle picking contributed to the final reconstruction. A workflow combining CryoSPARC[33] and RELION[34] was adopted for data processing, with the newly implemented data-driven regularization in RELION 5.0 beta[35] proving to be

crucial in reducing artifacts from preferred orientations and further enhancing the quality of the final maps (Supplementary Figs. 4 and 5).

Overall, the structures of the IFNλ4:IFNλR1:IL10Rβ-A3 and IFNλ3:IFNλR1:IL10Rβ-A3 complexes are similar to the previously solved ternary complex of an engineered, high-affinity IFNλ3 (H11) with IFNλR1 and IL10Rβ[27]. Most notably, the sequence differences between IFNλ3 and IFNλ4 lead to dramatic differences in the surface charge of the molecules which are readily visualized by coloring non-polar, basic, and positive residues differently (Supplementary Fig. 6). Viewing the ligand from the front, IFNλ3 has a large acidic patch that engages with both IFNλR1 and IL10Rβ, but mostly the latter; while the acidic regions of the protein contacting IFNλR1 are mostly conserved, this same region for IFNλ4 at the IL10Rβ interface contains more basic and neutral residues.

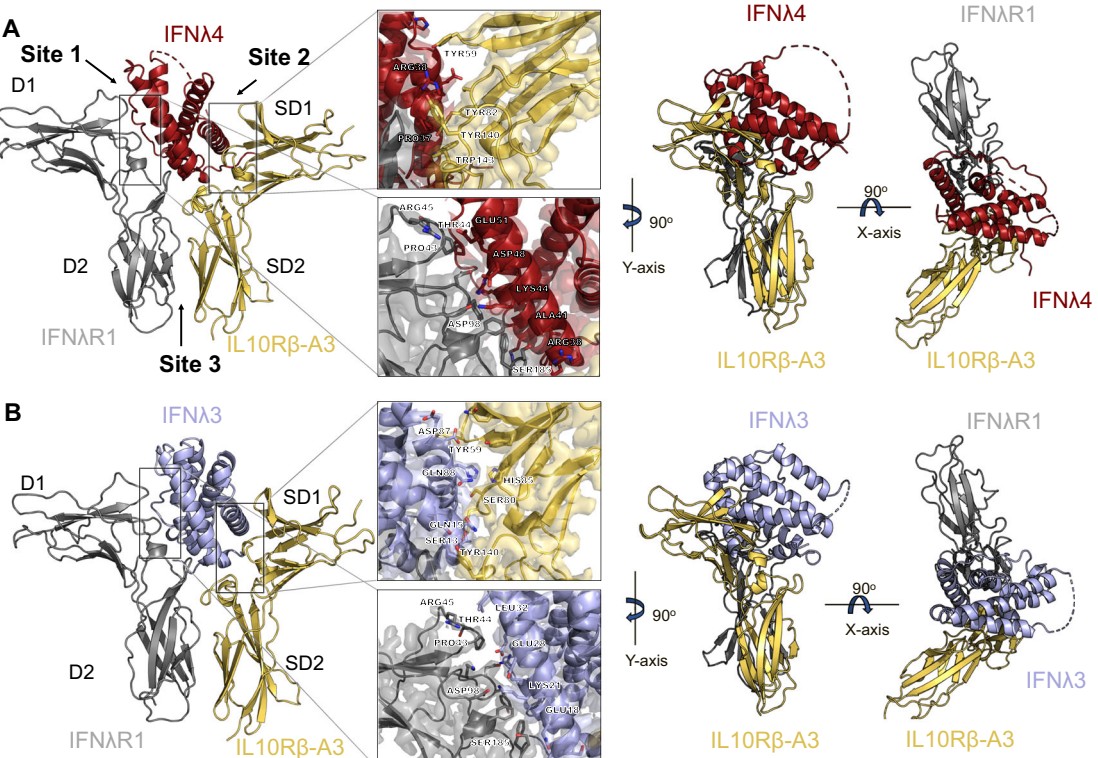

**Fig. 2 | Structures of the IFNλ4 and IFNλ3 ternary complexes. A** The structure of the IFNλ4 (red) ternary complex elucidates the mechanism of IFNλR1 (gray) and IL10Rβ-A3 (gold) engagement. IFNλ4 engages IFNλR1 with higher surface area and has fewer polar contacts with IL10Rβ. **B** The structure of the IFNλ3 (blue) ternary complex solved in parallel with the IFNλ4 complex provides a frame of reference for differences in receptor engagement of IFNλR1 and IL10Rβ-A3.

IFNλ3 and IFNλ4 make extensive contact with IFNλR1 and highlight the ability of the receptor to engage with basic residues, most notably Asp98 on IFNλR1 interacting with Arg47 of IFNλ4 and Lys21 of IFNλ3, which find themselves at similar locations and forming similar polar contacts at the interface (Supplementary Fig. 7A). However, IFNλ4 forms more polar contacts at this interface, with eight compared to four for IFNλ3, featuring unique involvement of the Ser185 residue on IFNλR1 forming three hydrogen bonds with residues Pro37, Arg38, and Ala41 (Supplementary Fig. 7B). In total, only 38 residues are conserved by the receptor binding domains of IFNλ3 and IFNλ4, with the greatest conservation occurring at the IFNλR1 interface on helices A and F (Supplementary Fig. 8). While the overall topology of this interface is similar between IFNλR1 and the IFNλs, these chains find themselves in closer proximity to IFNλR1 on IFNλ4, leading to a 38% higher buried surface area (BSA) at the IFNλ4:IFNλR1 interface (1740.7 Å$^2$) compared to IFNλ3:IFNλR1 (1260.1 Å$^2$) (Supplementary Fig. 9). This difference is driven by notable changes in the BSA of residues Pro43-Arg46 on IFNλR1 as well as differences in three conserved residues between the two proteins on IFNλ4 (Asp48, Glu51, and Leu55) experiencing large changes in individual BSA (50.34-fold, 2.34-fold, and 6.91-fold, respectively). Unique engagement with IFNλR1 of Arg65 (79.36 Å$^2$) and Pro66 (66.95 Å$^2$) on IFNλ4 have no analog on IFNλ3 (Supplementary Tables 1–3). Ultimately this leads to an affinity of 191 nM between IFNλ4 and IFNλR1 as measured by surface plasmon resonance (SPR) (Supplementary Fig. 10), more than 4-fold higher affinity than the previously published 850 nM interaction between IFNλ3 and IFNλR1[27]. The higher affinity of this interaction was also confirmed using microscale thermophoresis (MST) (Supplementary Fig. 11). Quantification of PPIs using molecular dynamics simulations reveals that IL10Rβ-A3 increases contact between IFNλ4:IL10Rβ-A3 as well as IFNλR1:IL10Rβ-A3 relative to the wild-type complex (Supplementary Fig. 12, Supplementary Tables 4 and 5). Further analysis of these simulations using dynamic cross-correlation generates insights

into protein complex behavior, revealing that the motions of the IFNλ4 complex are highly interdependent and that stronger anti-correlations are seen between IFNλ4 and IFNλR1 and IL10Rβ relative to IFNλ3 (Supplementary Fig. 13).

The importance of the hydrophobic residue network on IL10Rβ has been extensively described and consists of the Tyr59 residue in Loop 2, Tyr82 in Loop 3, and Tyr140 and Trp143 in Loop 5[27,36–39]. These residues indeed form extensive contacts with IFNλ4 and IFNλ3, but in distinct orientations (Fig. 3A, B). While Tyr140 remains in largely the same place, the Trp143 residue in IFNλ4 is in a flipped orientation compared to IFNλ3 but in both structures is buried at the interface. There is the same observed "pinch" on Helix A of IFNλ4 formed by Tyr140 and Trp143 as with IFNλ3-H11, with Tyr140 forming a hydrogen bond with Arg38 on IFNλ4 and Trp143 forming a hydrogen bond with the backbone oxygen of Leu35 (Supplementary Fig. 7C). Serendipitously, this Arg38 of IFNλ4 is in the same location as the Q15R mutation selected by directed evolution for IFNλ3-H11 and forms a similar hydrogen bond. Tyr82 engages spatially similar areas of IFNλ4 and IFNλ3 and in both cases is almost completely buried at the interface (>97%) but due to the differences in their surface charges, Tyr82 with IFNλ4 forms a hydrogen bond with the backbone nitrogen of Glu36 instead of the multiple hydrogen bonds formed with Ser11 and Ser13 on IFNλ3 (Supplementary Fig. 7D). Tyr59 in the IFNλ3 structure shares two hydrogen bonds with the ligand at Glu73 and Asp87, whereas there are no polar contacts shared with IFNλ4, indicating that the latter interaction is more hydrophobic in nature (Supplementary Fig. 7E). In total, IFNλ3 has four more polar contacts with IL10Rβ than IFNλ4.

As previously discussed, the overall topology of the IFNλR1 receptor is largely the same for the IFNλ4 and IFNλ3 complexes—as such, our general analysis is performed by aligning the two complexes using IFNλR1. This approach highlighted the previously described details of the IL10Rβ hydrophobic network but also revealed a significant rotation of the orientation IL10Rβ (Fig. 3C). This was calculated

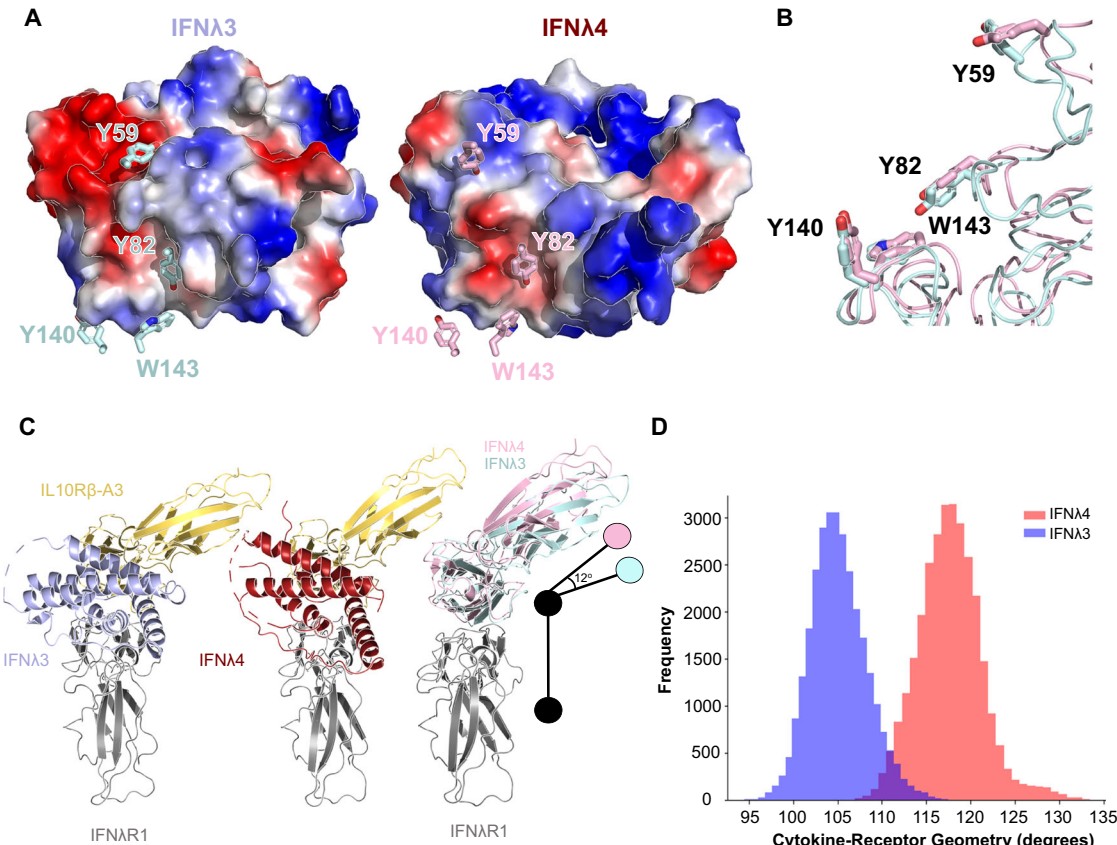

**Fig. 3 | The IFNλR1-IL10Rβ receptor complex engages IFNλ4 and IFNλ3 with distinct geometries. A** IFNλ3 (left) and IFNλ4 (right) both engage the hydrophobic residue network on IL10Rβ despite differences in surface charge and amino acid composition. **B** Exact differences in the orientations of Tyr59, Tyr82, Tyr140, and Trp143 on IL10Rβ when engaging IFNλ3 (blue) and IFNλ4 (red). **C** A distinct rotation of the IL10Rβ receptor is observed between the IFNλ3 (left) and IFNλ4 (middle) ternary complexes. 12-degree rotation is calculated after the alignment of the

IFNλR1 proteins (right) and calculating the angle between the center-of-masses in the protein complex. Protein complexes are overlayed and viewed from the top. **D** Histogram showing frequencies of cytokine-receptor geometry measurements throughout simulation time. Angle is calculated between the center-of-masses of the D1 domain of IFNλR1, IFNλ3 (blue), or IFNλ4 (red), and the SD1 domain of IL10Rβ-A3 as shown in 3C.

to be about 12° relative to the orientation of the IL10Rβ receptor for IFNλ3 by measuring the carbon alpha on the outermost residue (Ala46) of the outermost loop and calculating its position relative to the carbon alpha of Tyr82, which is in roughly the same spot for both complexes. This rotation takes place entirely in the SD1 domain of IL10Rβ and highlights the potential role of receptor geometry in both the recognition of diverse ligands as well as potentially explaining differences in experimental results observed between IFNλ3 and IFNλ4. Molecular simulations of the two complexes confirm that this difference in receptor geometry persists dynamically, with an average difference of 12.6° measured from the centers-of-mass of the D1 domain of IFNλR1, the respective IFNλs, and the SD1 domain of IL10Rβ-A3, and with the IFNλ4 complex experiencing a broader range of fluctuation (~25° for IFNλ4, ~20° for IFNλ3) (Fig. 3D, Supplementary Movies 1 and 2). Longer simulations show similar trends, with the IFNλ4 cytokine-receptor complex maintaining this difference in geometry (Supplementary Fig. 14). This rotation reflects the low sequence identity between the two ligands, leading to changes in receptor orientations.

The secondary structures of IFNλ3 and IFNλ4 are largely similar and the structure of IFNλ4 contains four helices that engage with its receptors (Fig. 4A). During structural refinement, however, we were unable to resolve the structure of Helix E of IFNλ4. While this region of the protein is uninvolved in receptor engagement, molecular modeling reveals that this highly basic region of IFNλ4 containing the amino acid sequence KRRHKPRR is predicted to lack secondary structure

unlike the corresponding region of IFNλ3, breaking from the structural symmetry of the two proteins. We sought to characterize the behavior of this region and used molecular dynamics simulations to observe behavioral differences between IFNλ4 and IFNλ3. By calculating the root mean squared fluctuation (RMSF) of these ligands while in complex with IFNλR1 and IL10Rβ throughout simulations, the increased disorder and flexibility of this region for IFNλ4 is quantifiable (Fig. 4B). More interestingly, throughout the course of these simulations, the negative ions in the simulation environment congregate within 1 Å exclusively around this region (Fig. 4C, Supplementary Fig. 15). Because of these observed behaviors in our simulations as well as the known tendency of IFNλ4 to induce cell stress and accumulate within the cell during protein translation and trafficking[17], we were curious to see if the removal of this region would have an impact on protein expression. We designed a chimeric version of IFNλ4 (xIFNλ4) by replacing Helix E of IFNλ4 with the corresponding region of IFNλ3 to see if there were observable differences in protein expression (Fig. 4D). Excitingly, we found that removing Helix E of IFNλ4 and replacing it with a more ordered and less basic peptide rescues protein expression in insect cells (Fig. 4E), suggesting this region may contribute to the poor secretion of IFNλ4.

Finally, with the ability to purify larger quantities of IFNλ4, we sought to characterize some of its in vitro functions and see if these structural differences would help explain differences in experimental results. We first measured in vitro phospho-STAT1 (pSTAT1) signaling in Hap1 cells relative to IFNλ3 as well as a type I IFN, IFNω1, which was

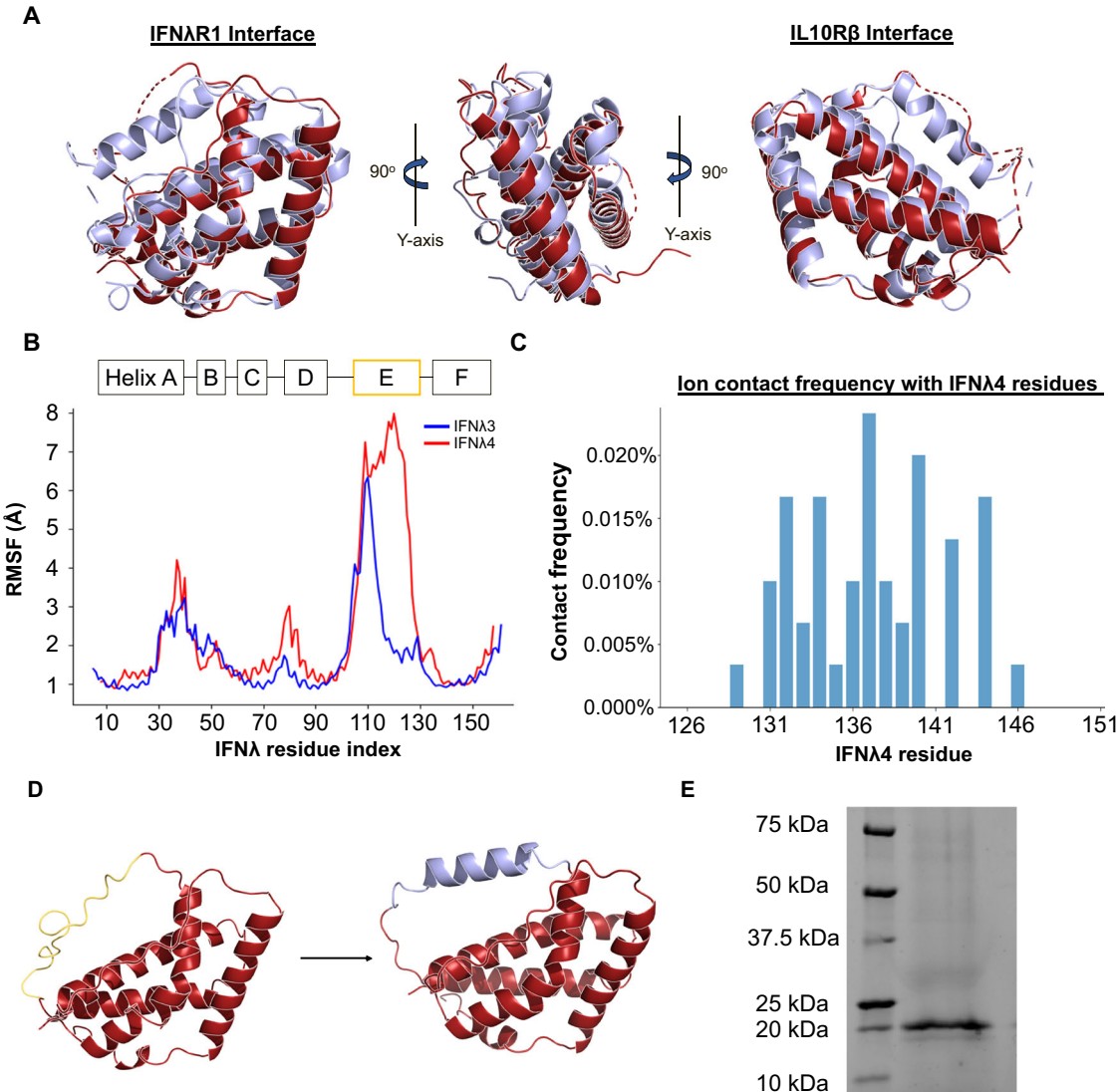

**Fig. 4 | Structural and dynamical analysis of IFNλ3 and IFNλ4 reveals a unique and disordered region of IFNλ4. A** Differences in secondary structure for IFNλ4 and IFNλ3 viewed from the IFNλR1 interface, the front of the ternary complex, and the IL10Rβ interface. Proteins are aligned to each other. **B** Root mean squared fluctuation (RMSF) of the IFNλ4 (red) and IFNλ3 (blue) proteins during molecular dynamics simulations reveals unique flexibility and disorder of Helix E. **C** Frequency of negative ion contact within 1 Å of Helix E on IFNλ4. **D** Design of a chimeric IFNλ4 (xIFNλ4) replacing the disordered and positively-charged Helix E (gold) with the corresponding region of IFNλ3 (blue). **E** SDS-PAGE protein gel of xIFNλ4. Image is representative of multiple independent experiments.

chosen as a positive control and further reference point due to its potency and similarities in function to type III IFNs (Fig. 5A). We confirm that treatment of cells with IFNλ4 activates the JAK/STAT pathway in a dose-dependent manner. Interestingly, we measured IFNλ4 to have a 20-fold lower EC50 than IFNλ3, while the EC50 IFNλ3 is 100-fold increased relative to IFNω1 (as previously reported[27]). This suggests that cells are more sensitive to IFNλ4 relative to IFNλ3 but that extracellular IFNλ4 signaling does not elicit the same potency as IFNλ3 as measured by Emax. These two properties of signaling (sensitivity and potency) are seen when A549 cells are treated with these IFNs as well (Supplementary Fig. 16), indicating that this differential behavior may reflect the structural differences reported above.

We next assessed the ability of IFNλ4 to induce interferon-stimulated genes (ISG) in Hap1 cells. Over short time frames (6 h), in keeping with the results of our pSTAT1 measurements, IFNλ4 potently induces the antiviral ISGs MX1 and ISG15 at saturating concentrations of protein, but not to the same extent as either IFNλ3 or IFNω1; further, IFNλ4 does not elicit a response from canonical anti-proliferative ISGs SAMD9L and APOL3 (Fig. 5B). Over longer time frames (24 h),

induction of ISG15 and MX1 was similar in Huh7.5.1 cells as well as primary human hepatocytes (PHH) at lower concentrations, however at higher concentrations of the protein, IFNλ3 and IFNλ4 elicit similar responses (Supplementary Fig. 17).

To further investigate how differences in signaling affect IFNλ function during Hepatitis C virus (HCV) infection, we next focused on changes in the expression of ISG15 and MX1 in infected Huh7.5.1 cells over time (Fig. 5C). From these results, we find a kinetic relationship between signaling and gene induction at both high and low concentrations of IFNλ, but with differences more notable at lower concentrations. At higher concentrations, we observe that IFNλ4 induces expression of MX1 and ISG15 more potently at shorter time scales, but that these differences drop off over time. This initial difference in potency at higher concentrations, however, leads to lower intracellular HCV genomic RNA levels in infected Huh7.5.1 cells, resulting in improved viral clearance following treatment with IFNλ4 at higher concentrations (Fig. 5D). Together, these results quantitatively confirm that IFNλ4 functions extracellularly as an antiviral protein and in some contexts can be more potent than IFNλ3, and that it is expression of the

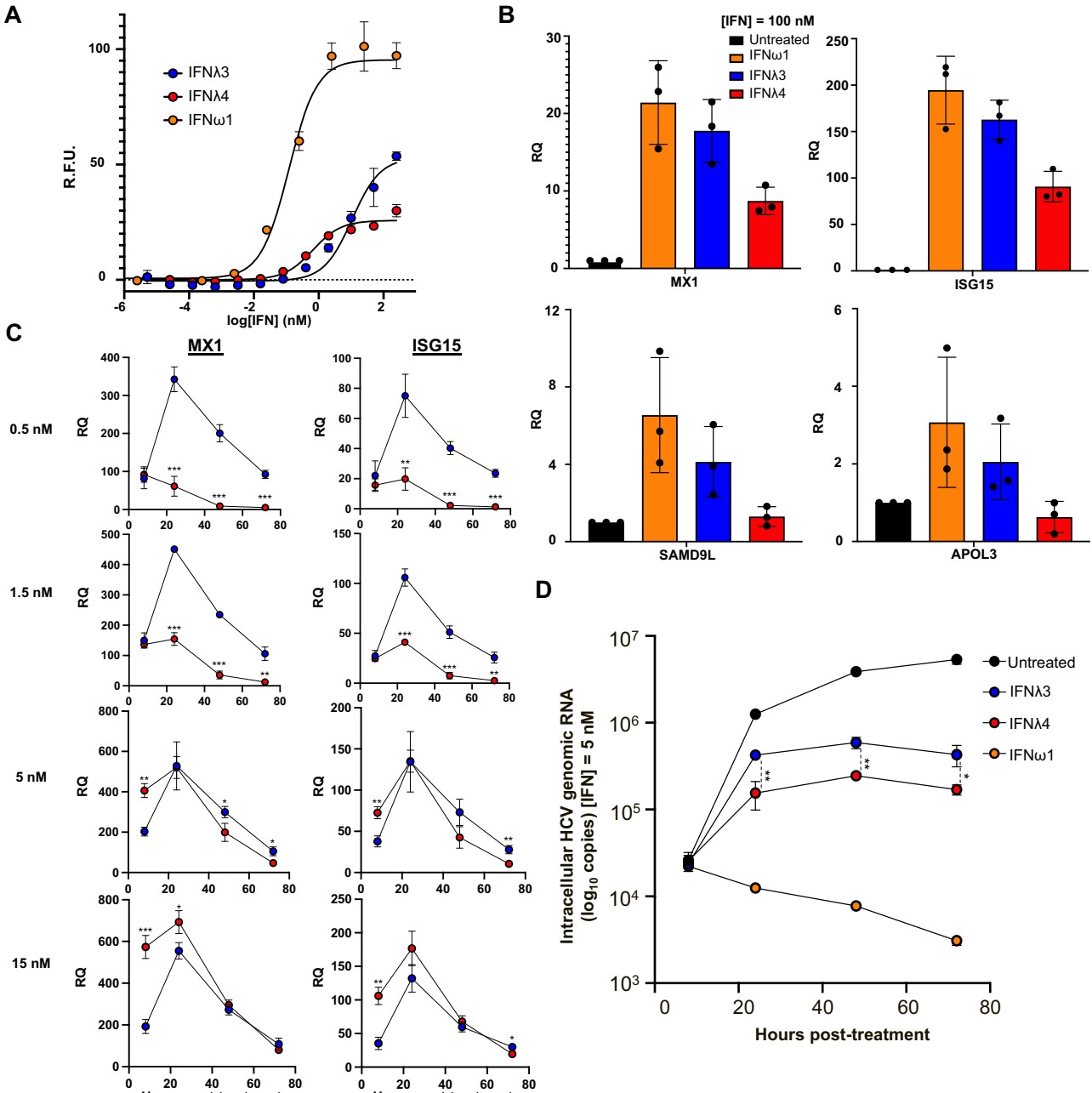

**Fig. 5 | IFNλ3 and IFNλ4 exhibit kinetic differences in antiviral gene induction, leading to differential antiviral activity in vitro. A** pSTAT1 signaling of IFNω1 (orange), IFNλ3 (blue), and IFNλ4 (red) in Hap1 cells. Curves are fit to a first-order logistic model. Data are presented with mean values and error bars representing ±SEM ($n = 3$ biologically independent experiments). **B** Relative quantification (RQ) of select genes induced by IFNω1, IFNλ3, and IFNλ4 in Hap1 cells treated with saturating concentrations (100 nM) interferon for 6 h as measured by qPCR. Data are presented with mean values. Error bars represent 95% confidence intervals ($n = 3$ biologically independent experiments). **C** Changes in induction of ISG15 or MX1 over time for IFNλ3 and IFNλ4 in Huh7.5.1 cells at a range of concentrations (-0.5, 1.5, 5, and 15 nM). Statistical significance determined by a two-tailed student $t$-test. Data are presented as mean values ± SD ($n = 3$ biologically independent experiments; * = $p \leq 0.05$, ** = $p \leq 0.01$, *** = $p \leq 0.001$). **D** Intracellular HCV genomic RNA level over time following treatment with IFNω1, IFNλ3, or IFNλ4 in Huh7.5.1 cells. Statistical significance was determined by a two-tailed student $t$-test. Data are presented as mean values ± SD ($n = 3$ biologically independent experiments; * = $p \leq 0.05$, ** = $p \leq 0.01$).

IFNλ4 protein and not its extracellular function that leads to decreased viral clearance in patients.

## Discussion

In this work, we present a robust method for the expression and purification of IFNλ4 that we use to solve the structure of the protein in complex with its two receptors, IFNλR1 and an affinity-matured IL10Rβ, as well as the ternary complex of IFNλ3 for reference. Through this

structural analysis, we identify key differences in ligand-receptor residue engagement and receptor geometry, that provide potential mechanistic explanations for differences in the in vitro function of IFNλ3 and IFNλ4. Using molecular modeling and simulation, we also identify and analyze an unstructured, highly positive region of IFNλ4 that prevents efficient secretion from cells as demonstrated by a chimeric IFNλ4, identifying a potential structural motif contributing to this behavior. To date, this study represents the most comprehensive

investigation of the IFNλ4 protein and provides solutions for long-standing issues surrounding IFNλ4 expression as well as questions regarding its structural and functional differences from IFNλ3[40].

The biochemical nature of IFNλ4 required significant effort to isolate the ternary complex for structural studies. Indeed, these 72 kDa structures represent some of the smallest protein complexes resolved by single-particle cryoEM to date, demonstrating the feasibility of single-particle cryoEM for other ternary cytokine-receptor complexes with only a small amount of purified complex. The structural insights gained from this study, most notably the distinct extracellular geometries and rotation of the IFNλR1-IL10Rβ receptor complex to accommodate such biochemically distinct ligands, build upon previous studies that have explored the role of receptor geometry and its connections to signaling amplitude, gene induction, and other key in vitro metrics of cell signaling[41–44]. Our results lend continued support to the importance of receptor geometry as a variable for therapeutic design and can be utilized as design considerations for future protein engineering and design efforts to signal through the tissue-specific type III IFN pathway. Our results also highlight the continued importance of experimentally determining structures of protein complexes, as our attempts to model the structure of the IFNλ4-IFNλR1-IL10Rβ complex using computational tools for structure prediction did not capture this difference in receptor geometry.

Our results quantitatively show that IFNλ3 and IFNλ4 elicit similar but distinct functions in their extracellular signaling abilities. While IFNλ4 has a lower EC50 for pSTAT1 signaling in line with its higher affinity for IFNλR1, it induces stronger initial ISG expression in HCV-infected cells only at higher concentrations relative to IFNλ3. It is expected that these increases in ISG induction during the early stages of infection would lead to increased clearance[45], but questions remain about how exactly IFNλ3 and IFNλ4 induce differential antiviral activity as well as the kinetics of signaling and gene induction that govern these differences. This in vitro functional data represents an important first step towards rigorous, quantitative characterization of IFNλ4 function, with comprehensive functional assays and exploration of the systems function of IFNλ4 representing future projects to better understand this protein. The robust and modular nature of our protein expression system also facilitates the characterization of clinically relevant IFNλ4 variants, as multiple point mutations have been identified in genetic studies as impactful for the in vivo function of IFNλ4[6,10,15,46–48].

The ultimate question about the role of IFNλ4 in the immune system and its systemic impact remains underexplored. Expression of IFNλ4 has been clearly linked to endoplasmic reticulum stress, leading to diminished viral clearance in patients with chronic HCV, and our results affirm that it is indeed expression, and not activity, of the IFNλ4 protein that leads to this behavior. The exact mechanism of why and how this stress is induced remains unresolved. We propose that the role of the disordered, highly basic Helix E of IFNλ4 is non-trivial and may lead to the recruitment of undesired interactions during protein translation and trafficking, leading to accumulation and associated stress. This hypothesis seems especially plausible in the context of other cytokines, such as IFNγ and IL-33, containing similarly charged and disordered regions that interact with components of the extracellular matrix for functions including sequestration[49–51]. To find one in the middle of a folded protein, though, is a curious and unique finding that is deserving of continued investigation. Cellular responses to the expression of xIFNλ4 would be an impactful next step towards understanding the structural basis of dysfunction in patients who express IFNλ4 and could motivate the design of small molecules[52] or protein binders to improve folding or stability of this chain, alleviating these differences in antiviral clearance. Whether IFNλ4 is meant to be retained in this fashion and that cell stress is indeed a desired outcome of the protein's expression, however, remains an important and unanswered question.

## Methods

### Chemicals, materials, and reagents
All chemicals and reagents were acquired from Thermo Fisher Scientific or MilliporeSigma unless otherwise specified.

### Molecular cloning and genetic constructs for protein expression
All relevant genetic constructs used in this study are provided in the supplementary information (Supplementary Table 6). All genes used for protein expression in this were either ordered from Twist Bioscience or were obtained from previously published genetic constructs[27,29]. A codon-optimized DNA sequence of the p179 isoform of IFNλ4 without amino acids 1–21 was used to reduce G:C content, facilitate amplification of the gene, and improve expression downstream. All DNA primers used for cloning, sequencing, and all other experiments were ordered from Sigma-Aldrich unless otherwise stated. PCR[53] was used in all cases to amplify genes and prepare them for molecular cloning, either by isothermal assembly[54] or using restriction digest for ligation[55]. Proteins expressed in mammalian systems utilized the pVLAD plasmid construct[56] and proteins expressed in insect systems utilized the pAC plasmid construct[30].

### Cell lines and cell culture
Baculovirus for protein expression was created using Sf9 cells (Gibco) cultured in Sf-900 II SFM media (Gibco). High Five (Hi5) insect cells (Expression Systems) for protein expression were cultured in Insect Xpress liquid media (Lonza) containing 1:1000 gentamycin sulfate (Corning) with gentle rotation at 28 °C. HEK293 GnTi- cells (ATCC) were cultured in FreeStyle 293 expression medium (Gibco) with 2% FBS and 1:1,000 penicillin-streptomycin (Gibco) with gentle rotation at 37 °C with 5% $CO_2$. HAP1 cells were a gift from Jan Carette at Stanford University and were cultured in DMEM with high glucose and no glutamine (Gibco) with 10% FBS and 1:1000 penicillin-streptomycin (P/S). Subculturing utilized 6-well plastic tissue culture plates and expansion utilized either T25 or T75 flasks at 37 °C with 5% $CO_2$. A549 cells were a gift from Curt Horvath at Northwestern University and were cultured using the same conditions. Huh7.5.1 cells are a derivative of the human hepatoma Huh7 cell line provided by F.V. Chisari of The Scripps Research Institute (La Jolla, CA) and were maintained under the same conditions as the HAP1 and A549 cell lines.

### Protein expression and purification
IL10Rβ-A3, IFNλ3, IFNω1, and IFNλR1 were expressed as previously described using baculovirus infection of either insect Hi5 (IL10Rβ-A3, IFNλ3, IFNω1) or HEK293 GnTi- (IFNλR1) cells[29]. During IFNλR1 expression, cells were moved to 30 °C. Buffer for all proteins was 1X HEPES-buffered saline (1X HBS, 20 mM HEPES + 150 mM NaCl, pH = 7.5) and samples were flash-frozen with 10% glycerol to protect samples before storage at −80 °C. Protein was quantified using UV-Vis and the presence of protein was determined using an SDS-PAGE Stain-Free protein gel (Bio-Rad).

High-yield expression of IFNλ4 is enabled by covalently linking the protein to its low-affinity receptor IL10Rβ, with sfGFP added as an additional chaperone protein to increase molecular weight and improve separation during size exclusion chromatography. After expression in insect Hi5 cells using the above protocol, the protein was harvested as previously described except with (1) a higher pH of the buffer, and (2) the addition of 500 mM NaCl at all steps (1X HBS + 500 mM NaCl, pH = 8.3). After harvest, the protein was treated with 3C protease to cleave IFNλ4 away from the sfGFP-10Rβ chaperone and the proteins were separated using an S75 SuperDex column (Cytiva). Protein fractions containing only IFNλ4 were then quantified via UV-Vis to determine the molar concentration and stored as above. xIFNλ4 was expressed and purified similarly except without 3C treatment.

## Yeast surface display and engineering of IL10Rβ

General yeast surface display protocols for quantification of protein binding and for directed evolution were performed as previously described[29,57]. To engineer IL10Rβ for higher affinity towards IFNλR1 and IFNλs, yeast displaying IL10Rβ were first stained with an anti-Myc antibody conjugated to Alexa 647 (Cell Signaling) to confirm protein display. Details on selection and engineering are presented in the results section of this manuscript. For yeast binding experiments, yeast displaying either IL10Rβ or IL10Rβ-A3 were stained to confirm protein display and normalize fluorescence for future data collection. Equimolar ratios of biotinylated IFNλR1 and either IFNλ3 or IFNλ4 were then titrated over a range of concentrations for 1 h at 4 °C with gentle rotation to determine relative bindings behaviors. Phosphate-buffered saline supplemented with 0.5% w/v bovine serum albumin (PBSA) was used to wash and remove unbound protein before resuspending yeast with 1:100 streptavidin tetramers conjugated to Alexa 647. Fluorescence readings were determined using flow cytometry. The data were then fit using a nonlinear regression in Prism (GraphPad version 9).

## Surface plasmon resonance

A Biacore T200 (GE Healthcare) at 22 °C was used to measure the binding affinities by the equilibrium method, as the kinetics of binding were too fast to measure. A Biotin CAPture chip and Biotin CAPture Reagent (50 µg/mL modified streptavidin in 0.01 M HEPES pH 7.4, 0.15 M NaCl, 3 mM EDTA, and 0.005 Surfactant P20) (Cytiva, ca. no. biotinylated 28920233) were used to reversibly capture biotinylated IFNλR1, with IFNλ4 as the analyte. For consistency, ~850 RU of biotinylated IFNλR1 was immobilized for each titration point. The running buffer was 10 mM MES pH 5.6, 500 mM NaCl, 10% glycerol, and 0.05% Tween20. The chip was regenerated using 6 M guanidine HCl and 0.25 M NaOH. The titration series was performed with nine points. The $K_D$ was calculated by fitting the equilibrium data using Biacore evaluation software and a 1:1 binding model in Prism (GraphPad version 9).

## Microscale thermophoresis to measure in-solution binding affinity

MST measurements were performed as previously described[58]. Briefly, to perform MST measurements, biotinylated IFNλR1 was labeled with fluorescent dye provided by NanoTemper (NT-L020). After confirming appropriate labeling, serial dilutions of IFNλ4:IFNλR1, IFNλ3:IFNλR1, or IFNλ4:IFNλR1:IL10Rβ were prepared in either 1X HBS (pH = 7.5) or 1X HBS (pH = 8.3) + 500 mM NaCl and analyzed using a NanoTemper Monolith X with technical assistance. Binding curves were fit using a 1:1 binding model in Prism (GraphPad version 9).

## In vitro pSTAT1 signaling assay

Hap1 cells were plated in a 96-well plate and treated with serial dilutions of either IFNλ4, IFNλ3, or IFNω1 for 15 min at 37 °C. After removing the supernatant, cells were then treated with trypsin (Gibco) for 5 min and resuspended gently. Following resuspension, cells were added to 16% PFA (Electron Microscopy Services) to fix for 10 min at room temperature before adding ice-cold methanol to permeabilize cells. Samples were stored at −80 °C until use. To quantify activation of pSTAT1, cells were washed thrice with 0.5% PBSA and then stained with an anti-Y701 pSTAT1 monoclonal antibody (Cell Signaling, product #9174S) according to manufacturer instructions. After incubation at 4 °C for at least 15 min with gentle rotation, samples were analyzed using a CytoFlex S flow cytometer (Beckman Coulter). Data were analyzed in Prism (GraphPad version 9).

## Quantification of gene induction by RT-qPCR

Initial quantification of gene induction was performed using Hap1 cells plated in a 6-well format (~600,000 cells) and treated with saturating concentrations IFNλ4, IFNλ3, or IFNω1 for 6 h before sample mRNA was harvested using a Monarch Total RNA miniprep kit T2010 (NEB)

following manufacturer instructions. cDNA was then synthesized via RT-PCR according to manufacturer instructions (Applied Biosystems) using 1 microgram of RNA. qPCR was performed using a QuantStudio Real-Time PCR system (Thermo Fisher Scientific) with SYBR Green as the reporter, again according to manufacturer instructions (Applied Biosystems). Transcription quantification was normalized to the expression of 18S ribosomal rRNA. Primers used for these experiments can be found in the supplementary information (Supplementary Table 7). Data were analyzed using the Relative Quantification (RQ) module available through Thermo Fisher Connect. RQ of ISG15 and MX1 in Huh7.5.1 cells and PHH was performed similarly, except RNA was extracted after 24 h and was normalized to GAPDH, using primers listed in the supplementary information (Supplementary Table 8).

## HCV infection and quantification of time-course gene induction by RT-qPCR

Huh7.5.1 cells were seeded in a 48-well plate at $4 \times 10^4$ cells per well. On the following day, the cells were infected with cell culture-derived HCV (HCVcc) virus (JFH1 strain) in a serum-free medium. After a 4-h adsorption period, infected cells were washed with phosphate-buffered saline and cultivated with complete media with appropriate concentrations of endotoxin-free IFNλ3, IFNλ4, or IFNω1 (~0.5, 1.5, 5, and 15 nM). HCV-infected cells were harvested at 8, 24, 48, and 72 h post-infection, and were used to extract RNA for real-time quantitative PCR (RT-qPCR). Total RNA was extracted using the Trizol reagent (Ambion) according to the manufacturer's instructions, and 30 ng of the total RNA was used per reaction. RT-qPCR analysis for ISG mRNA expression was performed using the Verso SYBR Green 1-Step qRT-PCR Low ROX kit (Thermo Scientific) on a QuantStudio 6 Pro Real-Time PCR System (Applied Biosystems). Primers used for this experiment can be found in the supplementary information (Supplementary Table 8). mRNA expression levels of human ISGs (ISG15 and MX1) were determined by the ΔΔCt method normalized to GAPDH mRNA. Intracellular HCV genomic RNA levels were quantified using a TaqMan probe specific for HCV 5′-UTR region via the Verso 1-Step qRT-PCR Low ROX kit (Thermo Scientific).

## Electron microscopy sample preparation

The IFNλ3 and IFNλ4 complexes were assembled by mixing purified IFNλ3 or IFNλ4, IFNλR1, and IL10Rβ-A3 at a 1:1:1.1 molar ratio. The mixture was incubated on ice for 30 min before being loaded onto a Superdex S75 column (Cytiva) equilibrated with either 1X HBS (pH = 8.3) + 500 mM NaCl for the IFNλ4 complex or 1X HBS (pH = 7.5) for the IFNλ3 complex. Peak fractions containing 0.1 mg/mL complex were taken directly off the column and used for grid preparation; it was crucial to avoid concentrating the protein complexes, especially IFNλ4, to prevent aggregation or loss by sticking to the concentrator membrane. Complexes treated with Endoglycosidase F and H were also tested to remove excess glycans on IFNλR1, however, it was found that this process resulted in severely aggregated samples. Sample vitrification was performed using a Vitrobot Mark IV (Thermo Fisher Scientific) operating at 22 °C and 100% humidity. A 3.5-µl sample was applied to holey carbon grids (UltrAuFoil 300 mesh Au 1.2/1.3) that had been glow-discharged for 30 s. The grids were blotted for 4 s at a "blotting force" 0 by standard Vitrobot filter paper (Ted Pella, product #47000-100) and were then plunge-frozen in liquid ethane.

## CryoEM data collection

Frozen grids were sent to the Advanced Electron Microscopy Facility at the University of Chicago for data collection. The dataset was acquired as movie stacks using EPU (Thermo Fisher Scientific) installed on a Titan Krios transmission electron microscope operating at 300 kV and equipped with a BioQuantum K3 imaging filter (Gatan). Images were recorded at a nominal magnification of ×81,000 and super-resolution counting mode by image shift. The total exposure time was set to 8 s

with 40 frames in a single stack and a total exposure of around 60 electrons $Å^{-2}$. The defocus range was set at −0.9 to −2.3 μm. Detailed parameters for cryoEM data collection are discussed in the supplementary information (Supplementary Table 9).

## CryoEM image processing

Raw images were imported to a CryoSPARC live session for motion correction, contrast transfer function (CTF) determination, and particle picking. Particles were automatically picked using blob picking (IFNλ4 dataset) or two-dimensional (2D) class averages generated from blob picking as templates (IFNλ3 dataset). The extracted particles were imported to CryoSPARC for further processing. To increase the number of good particles, Topaz train and Topaz extraction were also performed based on well-resolved 2D class averages. The extracted particles were then combined with particles from blob picking (IFNλ4) or template picking (IFNλ3). After additional 2D classification with non-overlapping particles, contamination and poorly aligned classes were discarded. The resulting particles were used to generate three initial models by ab initio reconstruction. Hetero refinement was then performed in CryoSPARC using the three initial models as the starting points. The coordinates of the particles from the best class were imported into RELION 5.0 beta[34] for particle re-extraction. Motion correction and CTF estimation were independently performed in RELION, with one round of both 2D and 3D classifications performed using the map generated from CryoSPARC as the initial map. RELION 5 Blush Regularization was applied throughout the process. The best class was subjected to 3D refinement, Bayesian polishing, CTF refinement, and post-processing. The final map of the IFNλ4 complex was resolved at 3.26 Å based on a standard acceptance criterion of a Fourier shell correlation (FSC) value of 0.143. The final map of the IFNλ3 complex was resolved at 3.00 Å to identical FSC standards.

## CryoEM model building, refinement, and validation

IFNλ4 model building was performed using a starting model of the IFNλ4 complex predicted by AlphaFold2[59]. Rigid body refinement was first performed in PHENIX[60] to adjust the relative orientation of the chains from the AlphaFold2 model, followed by real-space refinement in Coot[61]. The previously solved ternary structure of IFNλ3-H11/IFNλR1/IL10Rβ (PDB ID 5T5W) was used as the IFNλ3 starting model. The final model was refined in real space and validated using PHENIX. Molecular graphics were prepared using UCSF ChimeraX[62] and PyMOL v2.5[63]. Detailed statistics of model refinement and geometry are available in the supplementary information.

## Structure and sequence analysis

Structural alignments, models, and figures were generated in PyMOL. Sequence alignment of IFNλ4 and IFNλ3 was performed using Jalview[64]. BSA calculations as well as analysis of structural interfaces were performed using PISA[65].

## Molecular modeling and simulations

Molecular modeling and simulations were performed as previously described[29]. Briefly, classical MD simulations were performed in GROMACS 2020.5[66] with the AMBER ff99sb-ILDN* (Best 2009, Lindorff-Larsen 2010) force field[67,68]. Production runs were performed for sufficient time to reach equilibration as measured by RMSD (Supplementary Fig. 18), with 3 × 100 ns replicates for all simulations. Extensive details regarding system components and dimensions, simulation protocols, and technical parameters can be found in the supplementary information (Supplementary Tables 10–12).

The unresolved regions of the IFNλ4/IFNλR1/IL10Rβ-A3 complex were first modeled as informed by cryoEM data and completed in Chimera using AlphaFold2. A molecular model of the IFNλ4/IFNλR1/IL10Rβ complex was then generated from the structure of the IFNλ4/IFNλR1/IL10Rβ-A3 complex by editing the three engineered residues to

their wild-type amino acids using PyMOL, with the most probable rotamer conformations determined by lack of steric clashing and minimization of predicted strain from the PyMOL rotamer library.

## Residue contact analysis, root mean squared fluctuation calculations, dynamic receptor geometry calculations, dynamics cross-correlation analysis, and ion tracking

Residue contact analysis calculations were performed as previously described using CPPTRAJ[29,69]. For residue contact analysis, a cutoff distance of 10 Å was used for the quantification of PPIs. RMSF was calculated using MDAnalysis[70,71] with respect to the first frame of the production run in the simulation.

Angle measurements of the simulated protein complexes were calculated using the centers-of-mass of the D1 domain of IFNλR1, the respective IFNλs, and the SD1 domain of IL10Rβ-A3. Ion analysis was performed using the MDAnalysis Contact module. Production runs were analyzed by counting each frame in which any ion was within 1 Å of a residue's center of mass, expressed as a percentage calculated by dividing over the total number of frames.

All code relevant to these simulations and calculations can be found in the Supplementary Information, with further details available upon request.

## Reporting summary

Further information on research design is available in the Nature Portfolio Reporting Summary linked to this article.

## Data availability

All data relevant to the conclusions of this manuscript are included in the text, the supplementary information, or the source data file. The structural data, including coordinates and cryoEM maps, have been deposited in the Protein Data Bank (9BPU, 9BPV) and the Electron Microscopy Data Bank (EMD-44790, EMD-44791) with the corresponding accession codes. For any additional requests, please contact authors. Source data are provided with this paper.

## Code availability

Please visit https://github.com/bylehn/ifnl4-structure-paper for all computational resources and data relevant to this project. For any additional requests, please contact authors.

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

## Acknowledgements

This work was supported by funding from the Howard Hughes Medical Institute (HHMI), the Pritzker School of Molecular Engineering at the University of Chicago, and the National Institutes of Health (NIH) grants 1R35GM147179-01 to J.L.M. and 5R35GM143052-03 to M.Z. H.J., S.B.P., and T.J.L. are supported by the Intramural Research Program of the National Institute of Diabetes and Digestive and Kidney Diseases (NIDDK). H.J. is supported by a grant from the Korea Health Technology R&D Project through the Korea Health Industry Development Institute (KHIDI) funded by the Ministry of Health & Welfare, Republic of Korea (#RS-2023-00268767). The authors would like to acknowledge the University of Chicago Research Computing Center (RCC), which is supported in part by the NIH High-End Instrumentation (HEI) grant program (award S10OD028655), as well as the University of Chicago Advanced Electron Microscopy core facility for assistance. The authors acknowledge Ludmila Prokunina-Olsson and Olusegun Onabajo and thank them for their generous gifts of plasmid constructs containing IFNλ4, refolded IFNλ4 protein used for yeast selection, as well as many helpful conversations. We are grateful to Nathan Wallace with NanoTemper Technologies for technical assistance in performing MST measurements. Biorender was used in part to create Fig. 1A, B.

## Author contributions

W.S.G. and J.L.M. conceptualized the study. W.S.G. developed the protocol for IFNλ4 expression and purification, performed all yeast display and engineering, expressed and purified all proteins, designed the chimeric IFNλ4, and performed initial ISG measurements and signaling assays. B.Z. and M.Z. collected cryoEM data and refined the structure of the IFNλ4 and IFNλ3 protein complexes. W.S.G. and B.Z. analyzed structural data. W.S.G., A.K., and K.P. designed plasmid constructs for IFNλ4 expression and performed all molecular cloning. F.B. and W.S.G. performed molecular modeling and simulation and analyzed data with J.J.d.P. supervising the work. J.M.P. performed SPR and analyzed data with E.O. supervising the work. H.J. and S.B.P. performed antiviral assays and time-course gene induction assays with T.J.L. supervising the work. All authors contributed to the writing, reviewing, and editing of this manuscript.

## Competing interests

The authors declare no competing interests.
