## [Transparent Peer Review file · Nature Communications]

Structural studies of the IFN λ 4 receptor complex using cryoEM enabled by protein engineering

Corresponding Author: Dr Juan Mendoza

Version 0:

Reviewer comments:

Reviewer #1

(Remarks to the Author)

This structure-function study from Grubbed et al. provides insight into the signaling mechanism of the cytokine IFN λ 4. IFN λ 4 is an interferon variant with paradoxical behavior, in that its expression correlates with reduced viral clearance despite signaling normally in vitro. IFN λ 4 has been notoriously difficult to purify, and the authors solved this problem using an innovative strategy that involved expressing IFN λ 4 and its receptor IL10Rbeta as a single chain. They also engineered affinity-enhancing mutations in IL10R-beta so they could stabilize the formation of otherwise transient IFN λ 4/3-IL10Rbeta-IFN λ R1 complexes for cryoEM structure determination. The ternary complex structures revealed that IFN λ 4 is rotated by $\sim 12^\circ$ relative to the position of the more conventional homolog IFN λ 3, which agrees with molecular dynamics simulations. The structure also identified a disordered region of IFN λ 4 that reduced secretion levels and, presumably, the amount of IFN λ 4 available for receptor activation. The authors coupled their structural analysis with rigorous studies of IFN λ 4 signaling to investigate potency, E_{max} , target gene expression, and antiviral activity. They found that IFN λ 4 was more potent than IFN λ 3 but had a lower E_{max} , and that this signaling profile was associated with reduced expression of target genes. However, they found that higher (>100 ng/mL) amounts of cytokine reversed this effect, causing IFN λ 4 to induce greater levels of MX1 and ISG15 expression than IFN λ 3 at equivalent doses. This boost in antiviral gene expression was also associated with reduced HCV replication relative to IFN λ 3.

Collectively, this work answers several key questions about the molecular signaling of IFN λ 4. The major strengths are the creative methods employed for protein purification and structure determination, and the detailed analysis of the signaling properties of the disease-associated IFN λ 4 protein. The only notable weaknesses are some issues with clarity and semantics, as well as rotamer outliers in the structure that should be fixed prior to publication. This paper will appeal to a broad audience of immunologists, virologists, structural biologists, and those interested in cytokine/receptor signaling. If the items below are corrected, I strongly recommend its publication in Nature Communications.

Specific comments

- 1) Throughout the manuscript, there nomenclature issues that should be updated for clarity. In structural biology, "solution" is traditionally used to refer to solving the phase problem, which is not required for cryoEM. This should be replaced with "structure determination" or "structural studies" in the title.
- 2) Similarly, the authors often use the very broad term of engineering to mean affinity-enhancement or affinity-maturation (e.g. abstract line 25). Using either of these latter descriptions would be clearer.
- 3) In line 61, does 28% refer to sequence identity or similarity? And is this at the amino acid or DNA level? This should be clarified.
- 4) In line 111, if the increased NaCl in purification is mentioned, the concentration should be specified.
- 5) The use of yeast display to stabilize complexes for structure determination is innovative, especially in cryoEM where samples are often analyzed at concentrations well below their dissociation constants. The authors should cite other recent studies where this has been successful (IFN-gamma, IL-10, etc.).
- 6) In the PDB validation report, there are 22 rotamer outliers, which seems very high for a 3-angstrom structure. These should be mostly corrected before deposition.
- 7) In Figure 3D, it is unclear what the angle on the x-axis refers to. I am guessing that it refers to the relative angle between the two IFN-binding domains of the receptors? If so, this could be shown on 3C.
- 8) IFN omega1 is studied in the signaling assays in Figure 5, but it is not mentioned anywhere throughout the text. A brief explanation of what this IFN does and why it was chosen for comparison would be helpful for non-specialists.

9) In figure 8A, concentrations are in [nM] but in 8C IFN concentrations are in ng/mL. This makes it difficult to cross-compare concentrations between the dose-response curves in 8A and the plots below in 8C.

10) The paradox posed in the introduction is that IFN λ 4 is associated with lower viral clearance but retains activity in vitro. The authors' data support this in vitro activity, although viral clearance was significantly increased when using IFN λ 4 compared to IFN λ 3. In the discussion, the in vivo detrimental effects of IFN λ 4 are attributed to poor expression. Based on this assertion, do the authors speculate that reducing the amounts of IFN λ 4 and IFN λ 3 would reverse this effect in their HCV assay (meaning IFN λ 3 would be more effective).

11) The authors use 100ng/mL in Fig. 5D. Is this representative of a physiological concentration?

12) In lines 306-307, EC50 is said to be 100-fold different. The word different should be replaced with "increased" or "decreased" for clarity.

Reviewer #2

(Remarks to the Author)

The authors have to be congratulated for successfully producing significant amounts of IFN λ 4, and for resolving the structure of IFN λ 4 bound to its receptor. This reviewer is not a structural biologist and restricts his comments to the last part of the paper shown in figure 5 and supplementary figure 13.

The functional studies with purified IFN λ 3 and IFN λ 4 largely confirm previous publications: IFN λ 4 is a highly potent antiviral protein. The novelty comes from the availability of purified and well characterised IFN λ 4.

I would suggest to consistently use either nM or ng/mL for the concentration of IFN λ 3, IFN λ 4 in Figures 5A and 5C, and to indicate the concentration of the IFNs in Figure 5B and 5D also in the figure, not only in the text.

Reviewer #3

(Remarks to the Author)

Summary:

The paper describes a study focused on the IFN λ 4 protein, investigating its structural and functional differences from IFN λ 3. The authors have developed a method for high-yield expression and purification of unmodified IFN λ 4 to enable structural and functional studies. They have used protein engineering to overcome affinity limitations for structural determination, leading to the resolution of the 72 kDa structures of IFN λ 4 and IFN λ 3 in complex with IFN λ R1 and an engineered IL10R β using cryoEM. The study provides structural insights into the distinct extracellular geometries of the IFN λ 4 receptor complex and highlights the importance of receptor geometry in therapeutic design. The research also explores the biochemical nature of IFN λ 4 and its implications for cell signaling pathways and potential therapeutic interventions. Additional methods in this study include molecular cloning, genetic constructs, molecular modeling and the use of cell lines for protein expression.

My main objective is to assess the molecular dynamics (MD) simulations utilized in this study so my comments will address primarily these sections of the manuscript.

- The authors have utilized MD simulation as a tool to investigate the structural geometry of distinct receptors rather than as aid in assessing the dynamical features/associations of the receptor complexes studied in this work. In this capacity, they have utilized the structural information gained from the MD simulations as a means to interpret the possible functional differences of the receptors IFN λ 4 and IFN λ 3. Specifically, from the MD simulations performed and from the subsequent structural analyses conducted - they identify a disordered and non-receptor interacting region of IFN λ 4 that possibly identifies a structural explanation for the receptors low expression.
- After models were created, MD simulations were performed for 100 ns containing 3 distinct replicates of the receptor complex. Standard MD analysis tools were utilized to infer the receptor complex geometry and inter- and intra-protein interactions. In particular, the "most probable" starting structure of WT receptor used for the MD simulations was primarily determined based on the interaction database of the rotamer library of a molecular visualization program (PyMOL).
- In my opinion the approach for using MD simulation as a viable tool for assessing the structural interactions within a large receptor complex is a reasonable one. In fact, the authors used their initial results from the performed MD simulations to refine the experimental receptor expression. My main reservations are that based on the initial ambiguity of both inter-receptor interactions and particularly the angle rotamers of the distinct receptors within the complex - that too few replicates were run and for such a limited production run time (100 ns). Principally, there are a lot of unknowns and the authors rely heavily on the output of the simulations for a functional interpretation of the distinct receptors investigated. Therefore, if possible I would recommend conducting longer runs (200 – 300 ns) on a much larger number of distinct replicates (like 10 at least) for improved statistics and a more comprehensive understanding of the structural output, particularly since so much of the interpretation on the structural geometry (and in this case also the functional dynamics) of the receptors is based on the simulations. Further, since the details pertaining to the structured/un-structured regions of the receptor complex appear to be crucial in formulating a comprehensive understanding of the receptor signaling pathway(s) - a MD network analysis approach (such as dynamic cross-correlation analyses, dynamic residue interaction networks, etc..) should also be conducted to more extensively explore the relationship between conformational dynamics and flexibility in the receptors structure/function.

(Remarks to the Author)

In the present manuscript, authors have provided a production method for IFN λ 4 enabling structural studies of this IFN with its receptors and other experiments which were previously difficult due to its limited expression and purification. The protein production method will help community in their efforts to study IFN λ 4 in human immunology. Authors have further obtained cryo-EM structures of two IFN-receptor complexes adding to repertoire of small sized protein complexes solved by cryo-EM.

Comments:

1. *Lines 27-30: Comparison of the structures highlights differences in receptor engagement and reveals a distinct 12-degree rotation in overall receptor geometry, **providing a mechanistic explanation for differences in cell signaling, downstream gene induction, and antiviral activities.***

*Lines 87-91: Lastly, with our improved method for protein expression, we perform quantitative investigations of the extracellular function of IFN λ 4 and identify kinetic relationships between gene activation and viral clearance for the IFN λ 3 and IFN λ 4 proteins, **with these differences in vitro potentially explained by the structural differences in the receptor complexes.***

The above statements give the impression that the structural differences in the IFN λ 3 and IFN λ 4 receptor complexes are used to rationalize the differences observed in *in vitro* signaling experiments with a plausible mechanism contributing to the differences in signaling. However, structures or structural differences in the IFN λ 3 and IFN λ 4 receptor complexes are not discussed in the context of the differences in the *in vitro* signaling experiments when the results for these experiments are described later for various IFNs. As such the above statements seems misleading. Further, structural differences observed for structures studied outside the cell but explaining the differences in cell signaling, downstream gene regulation seems like an ambitious claim.

2. *Lines 159-160: To facilitate structural determination, an N-glycan minimized version of the IL10R β -A3 protein was used as previously described²⁶*

According to reference 26, N-glycan minimized version had four N-linked glycan sites which were mutated to Gln to facilitate crystallization studies which is understandable. However, Cryo-EM facilitates studies of glycosylated proteins relative to X-ray crystallography. Was there a specific reason to use the N-glycan minimized version of IL10R β -A3 protein as opposed to the glycosylated protein in Cryo-EM studies? The rationale is not clear in the manuscript. If the glycans are computationally modeled in the cryo-EM structure (e.g. using AF3 server), will that affect the interface/interpretation?

3. Significant text (Lines 180-241) has been devoted to describing the interface of the complexes including mentioning of specific residues and the type of interactions they make (e.g. Ser185 forming hydrogen bonds with Pro37, Arg38 etc.). However, no corresponding image/graphic has been provided either in the main text or in supplementary to follow the text. As such the description is hard to follow. I suggest authors to add corresponding images which describes the interactions at the interface discussed in the text.

4. Follow up to my above comment, please see Lines 223-241 with multiple mention of hydrogen bonding between residues at the interface of the receptor complexes e.g. "*Tyr82 with IFN λ 4 is more perpendicular to the ligand and forms a hydrogen bond with the backbone nitrogen of Glu36 instead of the multiple hydrogen bonds formed with Ser11 and Ser13 on IFN λ 3*"

Without proper images of the structure interfaces depicting what is written in Lines 223-241 or access to maps and models, it is difficult to review this information.

5. I suggest authors to add corresponding figures (main or supplementary) which describes the interface interactions mentioned in Lines 180-241. Further, the figures that describe detailed interactions should also include the EM density to support the modeled structure and detailed interactions.

6. *Line 234: ... Tyr82 with IFN λ 4 is **more perpendicular** to the ligand ...*

*Lines 236-237: Tyr59 displays extreme differences in orientation, with **significant rotation** ...*

*Line 240-241: In total, IFN λ 3 has **more polar contacts** with IL10R β than IFN λ 4*

I suggest authors to quantify terms like **more perpendicular, significant rotation and more polar contacts** here and at other places in the text.

7. *Supplemental Table 4 and 5: Protein-Protein interactions favored by a the engineered IFN λ 4 or wild type complex.*

It is not clear in what context the wild type and engineered terms are used here. Both, IL10R β and IFN λ 4 have wild type and engineered constructs described in the manuscript. If the tables are with respect to wild type IL10R β and engineered IL10R β -A3 then it should be appropriately mentioned in Protein 2 column in Tables 4 and 5. There is no mention of IL10R β -A3 in either of the tables.

8. The role of receptor geometry and the 12° orientational shift of IL10R β between IFN λ 3 and IFN λ 4 has been suggested to be important in manifesting the differences observed in experimental results between the two IFNs. Given that the authors

have structures for both IFN-receptor complexes, it can be commented as which interactions are broken and/or formed with 12° orientational shift between two IFN-receptor complexes.

9. *Supplemental table 1-3. Differences in BSA for residues of IFNλ3 and IFNλ4.*

Many of the residues used in these calculations have poor fit to the EM map according to the validation reports provided by the authors e.g. residues 55, 59, 63, 64, 65 for IFNλ4 putting into question the BSA calculations and their subsequent interpretations in the text. I suggest authors to ensure that the residues at the interface have proper density information for their modeling and proper stereochemistry (regarding side-chain outlier, clash etc) for reliable calculations including hydrogen bonds.

10. **Lines 223-241:** Much of this text can be shortened and summarized by providing a tabular format for hydrogen bonds including mention of donor and acceptor atoms. At present hydrogen bond mentions donor-acceptor atoms in certain residues and omits the information in other residues. Please also see comment 3 and 4.

11. Cryo-EM generally provides structural information on conformational heterogeneity in proteins and protein complexes possibly also providing conformations for distinct steps. Was there other conformationally distinct and relevant classes observed during cryo-EM data processing of the receptor complexes? Since the previous ternary complex was a crystal structure (PDB 5T5W), authors have unique opportunity to discuss this with access to cryo-EM data for both IFN complexes.

12. Since authors have already conducted MD simulations of the receptor complexes and have calculated protein-protein interactions from the trajectories, it can be specifically commented for interface interactions described in lines 180-241 that are maintained during the length of MD runs thereby possibly commenting on the dynamicity of the interaction interface.

Version 1:

Reviewer comments:

Reviewer #1

(Remarks to the Author)

The authors have thoroughly addressed all of my critiques. I congratulate them on their excellent manuscript.

Reviewer #2

(Remarks to the Author)

The authors have answered the points raised by me.

Reviewer #3

(Remarks to the Author)

The authors have addressed all of my questions and concerns. I recommend the manuscript for publication.

Reviewer #4

(Remarks to the Author)

The authors have addressed all of my suggestions and comments and have accordingly revised the manuscript adding additional figures, data and quantifications, improving clarity in structural analysis and descriptions in the text. I recommend publication of the revised manuscript.

October 14, 2024

Dear Editors:

Please find our response to reviewer's feedback on the manuscript now titled "**Structural studies of the IFN λ 4 receptor complex using cryoEM enabled by protein engineering**" by Grubbe *et al.* within this document. We are thankful for feedback provided through the review process improving the manuscript, and we are thankful for their positive and constructive feedback. We have included a thorough, itemized response to reviewer feedback in blue text, and changes within the manuscript are shown in blue text as well.

--

REVIEWER COMMENTS

Reviewer #1 (Remarks to the Author):

This structure-function study from Grubbed et al. provides insight into the signaling mechanism of the cytokine IFN λ 4 cytokine. IFN λ 4 is an interferon variant with paradoxical behavior, in that its expression correlates with reduced viral clearance despite signaling normally in vitro. IFN λ 4 has been notoriously difficult to purify, and the authors solved this problem using an innovative strategy that involved expressing IFN λ 4 and its receptor IL10R β as a single chain. They also engineered affinity-enhancing mutations in IL10R β so they could stabilize the formation of otherwise transient IFN λ 4/3-IL10R β -IFN λ R1 complexes for cryoEM structure determination. The ternary complex structures revealed that IFN λ 4 is rotated by $\sim 12^\circ$ relative to the position of the more conventional homolog IFN λ 3, which agrees with molecular dynamics simulations. The structure also identified a disordered region of IFN λ 4 that reduced secretion levels and, presumably, the amount of IFN λ 4 available for receptor activation. The authors coupled their structural analysis with rigorous studies of IFN λ 4 signaling to investigate potency, E_{max}, target gene expression, and antiviral activity. They found that IFN λ 4 was more potent than IFN λ 3 but had a lower E_{max}, and that this signaling profile was associated with reduced expression of target genes. However, they found that higher (>100 ng/mL) amounts of cytokine reversed this effect, causing IFN λ 4 to induce greater levels of MX1 and ISG15 expression than IFN λ 3 at equivalent doses. This boost in antiviral gene expression was also associated with reduced HCV replication relative to IFN λ 3.

Collectively, this work answers several key questions about the molecular signaling of IFN λ 4. The major strengths are the creative methods employed for protein purification and structure determination, and the detailed analysis of the signaling properties of the disease-associated IFN λ 4 protein. The only notable weaknesses are some issues with clarity and semantics, as well as rotamer outliers in the structure that should be fixed prior to publication. This paper will appeal to a broad audience of immunologists, virologists, structural biologists, and those interested in cytokine/receptor signaling. If the items below are corrected, I strongly recommend its publication in Nature Communications.

We appreciate the positive comments and thorough summary of the work presented in our manuscript. We look forward to addressing the issues with clarity and semantics, as well as the rotamer outliers in the structure, such that the study is suitable for publication in Nature Communications.

Specific comments

- 1) Throughout the manuscript, there nomenclature issues that should be updated for clarity. In structural biology, "solution" is traditionally used to refer to solving the phase problem, which is not required for cryoEM. This should be replaced with "structure determination" or "structural studies" in the title.

We have adjusted the title to reflect these differences. The new title of the manuscript is now “Structural studies of the IFN λ 4 receptor complex using cryoEM enabled by protein engineering”.

2) Similarly, the authors often use the very broad term of engineering to mean affinity-enhancement or affinity-maturation (e.g. abstract line 25). Using either of these latter descriptions would be clearer.

We have adjusted all uses of the term “engineering” to either “affinity-enhancement” or “affinity-maturation” in lines 24, 78, 138, and 341 to increase the clarity of our writing.

3) In line 61, does 28% refer to sequence identity or similarity? And is this at the amino acid or DNA level? This should be clarified.

We apologize for any miscommunication. In line 61 (now line 58-59), the 28% refers to sequence identity at the amino acid level. We have corrected this wording.

4) In line 111, if the increased NaCl in purification is mentioned, the concentration should be specified.

We have added the increased salt concentration values of 500 millimolar (mM) to what is now line 107.

5) The use of yeast display to stabilize complexes for structure determination is innovative, especially in cryoEM where samples are often analyzed at concentrations well below their dissociation constants. The authors should cite other recent studies where this has been successful (IFN-gamma, IL-10, etc.).

Recent cryoEM structures, such as IL10 and IL11, are at least twice the size of the complexes in our manuscript, which we hope the reviewers agree adds to the significance of our structures and methods. Nevertheless, we have added relevant citations for recent work concerning structural studies of cytokine-receptor complexes using cryoEM as well as structures that were obtained after affinity maturation of the complexes (reference number 25, 26, and 28).

6) In the PDB validation report, there are 22 rotamer outliers, which seems very high for a 3-angstrom structure. These should be mostly corrected before deposition.

The rotamer statistics have been improved through further refinement, and the PDB and validation report have been updated and are provided along with the revised manuscript.

7) In Figure 3D, it is unclear what the angle on the x-axis refers to. I am guessing that it refers to the relative angle between the two IFN-binding domains of the receptors? If so, this could be shown on 3C.

The angle on the x-axis refers to the relative angle between the center-of-masses of the D1 domain of IFN λ R1, IFN λ 3 (blue) or IFN λ 4 (red), and the SD1 domain of IL10R β -A3. We previously had this description in the Figure 3 legend but have added clarifying text to the figure in panel 3D as well as new marks to panel 3C to show how this angle is calculated.

8) IFN omega1 is studied in the signaling assays in Figure 5, but it is not mentioned anywhere throughout the text. A brief explanation of what this IFN does and why it was chosen for comparison would be helpful for non-specialists.

We have provided additional context for our use of IFN omega1 (IFN ω 1) in lines 305-308 to motivate the use of IFN ω 1 in our studies for non-specialist readers.

9) In figure 8A, concentrations are in [nM] but in 8C IFN concentrations are in ng/mL. This makes it difficult to cross-compare concentrations between the dose-response curves in 8A and the plots below in 8C.

We interpret this feedback to be relevant to Figure 5 and have adjusted the units in the figures such that they are the same across Figure 5 (nM).

10) The paradox posed in the introduction is that IFN λ 4 is associated with lower viral clearance but retains activity in vitro. The authors' data support this in vitro activity, although viral clearance was significantly increased when using IFN λ 4 compared to IFN λ 3. In the discussion, the in vivo detrimental effects of IFN λ 4 are attributed to poor expression. Based on this assertion, do the authors speculate that reducing the amounts of IFN λ 4 and IFN λ 3 would reverse this effect in their HCV assay (meaning IFN λ 3 would be more effective).

We appreciate this point of feedback and acknowledge that rigorous extracellular characterization of IFN λ 4 relative to the other IFN λ s, including IFN λ 3, remains an exciting area of work for this field. We would speculate that trends in antiviral data would be comparable to the gene induction data shown in Figure 5C, and that IFN λ 3 could be more effective at lower concentrations. Because we speculate that it is expression of IFN λ 4 that leads to these issues in vivo, the in vitro treatments with the protein represent an almost separate set of experiments that only assess the extracellular function of IFN λ 4, and that there may be other cellular or genetic mechanisms at play that cause this clearly observable difference.

11) The authors use 100ng/mL in Fig. 5D. Is this representative of a physiological concentration?

100 ng/mL (now reflected as 5 nM) was chosen because that was the concentration used for our experiments in Figure 5C where the team found IFN λ 4 being similar in gene induction activity to IFN λ 3. The range of concentrations chosen for 5C (10, 30, 100, and 300 ng/mL of IFN; now represented as approximately 0.5, 1.5, 5, and 15 nM) represent a standard titration range for extracellular treatment of IFNs that represents the higher and lower ends of their signaling activities.

12) In lines 306-307, EC50 is said to be 100-fold different. The word different should be replaced with "increased" or "decreased" for clarity.

We have clarified the terminology used in now line 310 to communicate that the EC50 of IFN λ 3 is 100-fold higher than IFN ω 1, replacing the term "different" with "increased".

Reviewer #2 (Remarks to the Author):

The authors have to be congratulated for successfully producing significant amounts of IFN λ 4, and for resolving the structure of IFN λ 4 bound to its receptor. This reviewer is not a structural biologist and restricts his comments to the last part of the paper shown in figure 5 and supplementary figure 13.

We are deeply appreciative of the positive feedback provided by Reviewer #2 and look forward to addressing any comments or concerns.

The functional studies with purified IFN λ 3 and IFN λ 4 largely confirm previous publications: IFN λ 4 is a highly potent antiviral protein. The novelty comes from the availability of purified and well characterised IFN λ 4.

I would suggest to consistently use either nM or ng/mL for the concentration of IFN λ 3, IFN λ 4 in Figures 5A and 5C, and to indicate the concentration of the IFNs in Figure 5B and 5D also in the figure, not only in the text.

In line with requests from Reviewer #1, we have adjusted the concentration units to all be the same (nM) and have added new labels to Figure 5B and 5D.

Reviewer #3 (Remarks to the Author):

Summary:

The paper describes a study focused on the IFN λ 4 protein, investigating its structural and functional differences from IFN λ 3. The authors have developed a method for high-yield expression and purification of unmodified IFN λ 4 to enable structural and functional studies. They have used protein engineering to overcome affinity limitations for structural determination, leading to the resolution of the 72 kDa structures of IFN λ 4 and IFN λ 3 in complex with IFN λ R1 and an engineered IL10R β using cryoEM. The study provides structural insights into the distinct extracellular geometries of the IFN λ 4 receptor complex and highlights the importance of receptor geometry in therapeutic design. The research also explores the biochemical nature of IFN λ 4 and its implications for cell signaling pathways and potential therapeutic interventions. Additional methods in this study include molecular cloning, genetic constructs, molecular modeling and the use of cell lines for protein expression.

My main objective is to assess the molecular dynamics (MD) simulations utilized in this study so my comments will address primarily these sections of the manuscript.

- The authors have utilized MD simulation as a tool to investigate the structural geometry of distinct receptors rather than as aid in assessing the dynamical features/associations of the receptor complexes studied in this work. In this capacity, they have utilized the structural information gained from the MD simulations as a means to interpret the possible functional differences of the receptors IFN λ 4 and IFN λ 3. Specifically, from the MD simulations performed and from the subsequent structural analyses conducted - they identify a disordered and non-receptor interacting region of IFN λ 4 that possibly identifies a structural explanation for the receptors low expression.

This summary is largely correct, but it is important to define that neither IFN λ 4 nor IFN λ 3 is a receptor, and that we do not speculate on the expression of either of the two receptors that IFN λ 4 interacts with (IFN λ R1 and IL10R β). IFN λ 4 and IFN λ 3 are ligands for IFN λ R1 and IL10R β and understanding that insight is key to understanding the motivation for performing MD simulations on the cytokine-receptor complexes presented in this manuscript. Indeed, our simulations highlight the behavior of a “disordered and non-receptor interacting region of IFN λ 4”

- After models were created, MD simulations were performed for 100 ns containing 3 distinct replicates of the receptor complex. Standard MD analysis tools were utilized to infer the receptor complex geometry and inter- and intra-protein interactions. In particular, the “most probable” starting structure of WT receptor used for the MD simulations was primarily determined based on the interaction database of the rotamer library of a molecular visualization program (PyMOL).

To clarify, the “models” used for the MD simulations were the structures from our cryoEM studies of the IFN λ 3 and IFN λ 4 receptor complexes presented in this manuscript. We elected to use MD simulations to better understand the dynamics of the unresolved region of the IFN λ 4 protein in complex with its receptors. It is important to note that none of the residues in the unresolved regions of the IFN λ 3 and IFN λ 4 receptor complexes that were computationally modeled for simulations exist at protein interfaces.

We ran additional simulations of the IFN λ 4 complex with a computational model of the wild-type IL10R β receptor and used PyMOL to determine the most probable rotamer conformations of only the three residues that were engineered in the structure of IFN λ 4 determined by cryoEM. This was done to lend insights into differences in protein-protein interactions (PPIs) between the affinity-matured and wild-type IL10R β protein (see Grubbe 2023 BiophysJ). The MD analysis tools alluded to above were not used to “infer the receptor complex geometry”, as these values are accessible without simulations, and the quantification of inter- and intra-protein interactions were limited only to the comparison of the IFN λ 4 complex with the two previously mentioned forms of IL10R β .

• In my opinion the approach for using MD simulation as a viable tool for assessing the structural interactions within a large receptor complex is a reasonable one. In fact, the authors used their initial results from the performed MD simulations to refine the experimental receptor expression.

We appreciate that Reviewer #3 believes that our approach for studying cytokine-receptor complexes in this way is a “reasonable one”. We interpret the final line of the reviewer - “In fact, the authors used their initial results from the performed MD simulations to refine the experimental receptor expression” - to refer to the design of the chimeric IFN λ 4 (xIFN λ 4). xIFN λ 4 was designed using multiple insights generated from this manuscript as well as an experimental understanding of IFN λ 4 expression and purification, including the sequence and charge dissimilarity of these regions in IFN λ 3 and IFN λ 4, the inability to resolve this region of IFN λ 4 using cryoEM, and the difficulty isolating IFN λ 4 in a lab. The quantification of flexibility (RMSF) using MD simulations and the aggregation of ions around this charged and disordered region throughout MD simulation time indeed contributed to our design process, but were not the sole motivators for our design of xIFN λ 4. This is described in lines 270-290.

My main reservations are that based on the initial ambiguity of both inter-receptor interactions and particularly the angle rotamers of the distinct receptors within the complex - that too few replicates were run and for such a limited production run time (100 ns). Principally, there are a lot of unknowns and the authors rely heavily on the output of the simulations for a functional interpretation of the distinct receptors investigated. Therefore, if possible I would recommend conducting longer runs (200 – 300 ns) on a much larger number of distinct replicates (like 10 at least) for improved statistics and a more comprehensive understanding of the structural output, particularly since so much of the interpretation on the structural geometry (and in this case also the functional dynamics) of the receptors is based on the simulations.

We appreciate the feedback provided by this reviewer relevant to the simulation portion of this manuscript. Simulations were run using the structures of these complexes to observe dynamic differences in behavior and elevate our understanding of the static structures, particularly contrasting IFN λ 3 and IFN λ 4, not so much to “[interpret]...the structural geometry” or “functional dynamics of the receptors”. The structures themselves show differences in geometry and key interactions, many of which are detailed in the discussion of the structures in lines 174-259.

We do, however, respectfully disagree with the statement that there is “initial ambiguity of both inter-receptor interactions and particularly the angle rotamers of the distinct receptors within the complex”. The protein structures were solved using cryoEM, and we have since improved the already high-resolution and confidence of these structures (see feedback for Reviewer #1, updated deposition forms). Further, we believe that there is no more initial ambiguity than in any standard molecular dynamics simulations involving an atomic structure of a protein complex. Nevertheless, we appreciate that more simulation data unequivocally improves the statistics and our following analysis, and happily completed 3x300 ns new simulations for both the IFN λ 3 and IFN λ 4 receptor complexes presented in this manuscript. The differences observed between the 100 ns and 300 ns sets do not change our conclusions derived from the original simulations regarding persistence of receptor geometry differences (Figure 3D), but we have included this analysis as a new Supplemental Figure 14 and have elaborated on these results in lines 256-257. We do not believe that additional replicates will change our conclusions.

Further, since the details pertaining to the structured/un-structured regions of the receptor complex appear to be crucial in formulating a comprehensive understanding of the receptor signaling pathway(s) - a MD network analysis approach (such as dynamic cross-correlation analyses, dynamic residue interaction networks, etc..) should also be conducted to more extensively explore the relationship between conformational dynamics and flexibility in the receptors structure/function.

To clarify, the regions that were not resolved on IFN λ 4 during cryoEM and structure refinement do not engage with the receptors IFN λ R1 and IL10R β on the cell surface. Further, the signaling complex of IFN λ 3 is relatively well understood and has been previously reported (Mendoza 2017 Immunity). We agree, though, that the proposed method of analysis

is valid and will strengthen our understanding of the dynamics of these systems. We performed a dynamic cross-correlation (DCC) analysis on the simulation data to better understand the relationships between conformational dynamics and flexibility within the protein complex. In this analysis, we find that the motions of the IFN λ 4 complex are highly interdependent, and that stronger anti-correlations are seen between IFN λ 4 and IFN λ R1 and IL10R β relative to IFN λ 3. In turn, the IFN λ 3 complex shows less interdependence. These findings corroborate the differences observed in the affinities of the two ligands for the IFN λ R1 receptor, as well as the differences in polar and non-polar contacts between the complexes as described in the manuscript. We have added Supplemental Figure 13 and have communicated these findings in lines 206-210, as well as adding the raw DCC data to our supplementary materials.

Reviewer #4 (Remarks to the Author):

In the present manuscript, authors have provided a production method for IFN λ 4 enabling structural studies of this IFN with its receptors and other experiments which were previously difficult due to its limited expression and purification. The protein production method will help community in their efforts to study IFN λ 4 in human immunology. Authors have further obtained cryo-EM structures of two IFN-receptor complexes adding to repertoire of small sized protein complexes solved by cryo-EM.

We are grateful for Reviewer #4's insightful feedback on the structural work presented here and look forward to addressing their concerns and improving the manuscript as suggested.

Comments:

1.Lines 27-30: Comparison of the structures highlights differences in receptor engagement and reveals a distinct 12-degree rotation in overall receptor geometry, providing a mechanistic explanation for differences in cell signaling, downstream gene induction, and antiviral activities.

Lines 87-91: Lastly, with our improved method for protein expression, we perform quantitative investigations of the extracellular function of IFN λ 4 and identify kinetic relationships between gene activation and viral clearance for the IFN λ 3 and IFN λ 4 proteins, with these differences *in vitro* potentially explained by the structural differences in the receptor complexes.

The above statements give the impression that the structural differences in the IFN λ 3 and IFN λ 4 receptor complexes are used to rationalize the differences observed in *in vitro* signaling experiments with a plausible mechanism contributing to the differences in signaling. However, structures or structural differences in the IFN λ 3 and IFN λ 4 receptor complexes are not discussed in the context of the differences in the *in vitro* signaling experiments when the results for these experiments are described later for various IFNs. As such the above statements seems misleading. Further, structural differences observed for structures studied outside the cell but explaining the differences in cell signaling, downstream gene regulation seems like an ambitious claim.

We appreciate this feedback from Reviewer #4, and certainly did not intend to misrepresent any information contained within this manuscript. We believe that our findings - that a cytokine-receptor complex consisting of the same two receptors (IFN λ R1 and IL10R β) recognizes related ligands (IFN λ 3 vs IFN λ 4) with such distinct complex geometries (Figure 2, 3) as well as differences in cell signaling, gene induction, and antiviral activity (Figure 5) - provides strong data in support of the hypothesis, and consistent with other studies (references 41-44) that receptor complex geometry can influence *in vitro* activities of protein ligands. Nevertheless, we have modified the language contained in the abstract (line 28) and have elaborated on the potential links between receptor complex geometry and downstream cell signaling in the discussion (line 354-361). We do not desire to make a claim that receptor geometry alone is the only differentiating factor for the activities of IFN λ 3 and IFN λ 4, and had previously outlined our reasons for this choice. We hope that this explanation, as well as the above changes to the manuscript, reflect our acknowledgement of the insights from Reviewer #4.

2.Lines 159-160: To facilitate structural determination, an N-glycan minimized version of the IL10R β -A3 protein was used as previously described²⁶

According to reference 26, N-glycan minimized version had four N-linked glycan sites which were mutated to Gln to facilitate crystallization studies which is understandable. However, Cryo-EM facilitates studies of glycosylated proteins relative to X-ray crystallography. Was there a specific reason to use the N-glycan minimized version of IL10R β -A3 protein as opposed to the glycosylated protein in Cryo-EM studies? The rationale is not clear in the manuscript. If the glycans are computationally modeled in the cryo-EM structure (e.g. using AF3 server), will that affect the interface/interpretation?

Resolving the IFN λ 4 receptor complex took multiple rounds, and our initial attempts at resolving the structure led to heavily aggregated products and an inability to resolve independent particles. Of the many variables we changed, glycosylation of IL10R β -A3 was one of them. Initially, we had aimed to use X-ray crystallography for our structural studies as with Mendoza et. al 2017, motivating the use of the N-glycan minimized IL10R β . However, due to the biochemical behavior of IFN λ 4, the complex could not be concentrated to the range needed for crystallization, thus shifting our efforts to cryoEM. The glycans on IL10R β are not located at any interface of the protein complex and should not affect interpretation of the structural results presented in this manuscript (see Mendoza 2017). Additionally, single-particle selection was most successful with the N-glycan minimized version of IL10R β -A3.

3. Significant text (Lines 180-241) has been devoted to describing the interface of the complexes including mentioning of specific residues and the type of interactions they make (e.g. Ser185 forming hydrogen bonds with Pro37, Arg38 etc.). However, no corresponding image/graphic has been provided either in the main text or in supplementary to follow the text. As such the description is hard to follow. I suggest authors to add corresponding images which describes the interactions at the interface discussed in the text.

We thank the reviewer for this valuable suggestion. In response, we have revised Figure 2 to include detailed information on the interface between the ligand and receptor, along with corresponding cryoEM density. Additionally, we have created new Supplemental Figures 7 and 9 to illustrate the detailed interactions at this interface.

4.Follow up to my above comment, please see Lines 223-241 with multiple mention of hydrogen bonding between residues at the interface of the receptor complexes e.g. *“Tyr82 with IFN λ 4 is more perpendicular to the ligand and forms a hydrogen bond with the backbone nitrogen of Glu36 instead of the multiple hydrogen bonds formed with Ser11 and Ser13 on IFN λ 3”*

Without proper images of the structure interfaces depicting what is written in Lines 223-241 or access to maps and models, it is difficult to review this information.

We apologize for this inconvenience. We have modified Figure 2 to make it easier to interpret these results, as well as the new Supplemental Figures 7 and 9 to show the specific residues and corresponding maps. Further, we have included the digital files for these models and maps, and welcome the reviewer to access and interpret that information as they see fit.

5. I suggest authors to add corresponding figures (main or supplementary) which describes the interface interactions mentioned in Lines 180-241. Further, the figures that describe detailed interactions should also include the EM density to support the modeled structure and detailed interactions.

As mentioned above, we have updated Figure 2 and created Supplemental Figures 7 and 9 to illustrate the detailed interactions, specific residues and corresponding maps.

6. Line 234: ... Tyr82 with IFN λ 4 is more perpendicular to the ligand ...
Lines 236-237: Tyr59 displays extreme differences in orientation, with significant rotation ...
Line 240-241: In total, IFN λ 3 has more polar contacts with IL10R β than IFN λ 4

I suggest authors to quantify terms like more perpendicular, significant rotation and more polar contacts here and at other places in the text.

We have added the requested details and modified the language of the manuscript in lines 234-240.

7. Supplemental Table 4 and 5: Protein-Protein interactions favored by a the engineered IFN λ 4 or wild type complex.

It is not clear in what context the wild type and engineered terms are used here. Both, IL10R β and IFN λ 4 have wild type and engineered constructs described in the manuscript. If the tables are with respect to wild type IL10R β and engineered IL10R β -A3 then it should be appropriately mentioned in Protein 2 column in Tables 4 and 5. There is no mention of IL10R β -A3 in either of the tables.

We appreciate this note and understand where confusion may have arisen. These supplemental figures compare molecular dynamics simulations of the IFN λ 4/IFN λ R1/IL10R β -A3 complex presented in the manuscript to a model of the complex containing a wild-type IL10R β . We have adjusted the titles of Supplemental Tables 4 and 5 to better reflect the contents of the tables and have changed the names of the proteins in the Protein 2 column accordingly.

8. The role of receptor geometry and the 12° orientational shift of IL10R β between IFN λ 3 and IFN λ 4 has been suggested to be important in manifesting the differences observed in experimental results between the two IFNs. Given that the authors have structures for both IFN-receptor complexes, it can be commented as which interactions are broken and/or formed with 12° orientational shift between two IFN-receptor complexes.

We provide explicit details for differences in interactions in lines 223-259, including major differences in the hydrophobic binding network of IL10R β . We have also supplemented this pre-existing information with the new supplemental figures requested above.

9. Supplemental table 1-3. Differences in BSA for residues of IFN λ 3 and IFN λ 4.

Many of the residues used in these calculations have poor fit to the EM map according to the validation reports provided by the authors e.g. residues 55, 59, 63, 64, 65 for IFN λ 4 putting into question the BSA calculations and their subsequent interpretations in the text. I suggest authors to ensure that the residues at the interface have proper density information for their modeling and proper stereochemistry (regarding side-chain outlier, clash etc) for reliable calculations including hydrogen bonds.

We would like to note that the PDB validation was based on a higher contour level (5.2 σ) that we input when depositing the structure. IFN λ 4 exhibits reasonable, albeit relatively weaker, density compared to the receptors. To address this, we have created Supplemental Figure 9, which displays the fit of the corresponding residues at different contour levels. We have updated the values in these tables and manuscript accordingly.

10. Lines 223-241: Much of this text can be shortened and summarized by providing a tabular format for hydrogen bonds including mention of donor and acceptor atoms. At present hydrogen bond mentions donor-acceptor atoms in certain residues and omits the information in other residues. Please also see comment 3 and 4.

We appreciate this suggestion from the reviewer. We wish to retain the description in the text so that the detailed difference at the IL10R β interface can be illustrated. In the meanwhile, following the reviewer's suggestions in

comments 3 and 4, we have included Supplemental Figure 9 to visually depict H-bond interactions described in the text.

11. Cryo-EM generally provides structural information on conformational heterogeneity in proteins and protein complexes possibly also providing conformations for distinct steps. Was there other conformationally distinct and relevant classes observed during cryo-EM data processing of the receptor complexes? Since the previous ternary complex was a crystal structure (PDB 5T5W), authors have unique opportunity to discuss this with access to cryo-EM data for both IFN complexes.

We thank the reviewer for this insightful question. We actually did not observe any conformational heterogeneity in either the IFN λ 3 or IFN λ 4 datasets. It is important to note that we performed extensive particle picking, 2D classification, and 3D classification ($k = 3$, iter = 150, Supplemental Figure 4 and 5) to obtain the best subsets of particles that could be reconstructed to higher resolution. While we cannot entirely rule out the possibility that conformationally distinct particles may have been discarded during these processes, the absence of noticeable classes with different conformations in the 3D classification suggests that the subset we obtained is relatively homogeneous. As discussed in the text, there is a significant conformational difference between the IFN λ 3 and IFN λ 4 complexes. The structure of the IFN λ 3 complex agrees with the crystal structure (main chain RMSD=0.609), indicating that the differences are likely due to the different ligands present in the complexes.

12. Since authors have already conducted MD simulations of the receptor complexes and have calculated protein-protein interactions from the trajectories, it can be specifically commented for interface interactions described in lines 180-241 that are maintained during the length of MD runs thereby possibly commenting on the dynamicity of the interaction interface.

We have included a new dynamics cross-correlation analysis to better investigate differences in interface interaction. Please see the detailed responses to Reviewer #3 for more information on our findings and the availability of our data.

--

Please let us know if there are any outstanding points of feedback and we will gladly address them.

Sincerely,

Juan L. Mendoza, Ph.D.

Assistant Professor of Molecular Engineering
Pritzker School of Molecular Engineering and
Dept. of Biochemistry and Molecular Biology

Freeman Hraboski Scholar, Howard Hughes Medical Institute
The University of Chicago